# Belief Dynamics Reveal the Dual Nature of In-Context Learning and Activation Steering

**Eric Bigelow** [* 1 2 3]  **Daniel Wurgaft** [* 1 4]  **YingQiao Wang** [2]
**Noah Goodman** [4 5]  **Tomer Ullman** [2 6]  **Hidenori Tanaka** [3 6]  **Ekdeep Singh Lubana** [1]

## Abstract

Large language models (LLMs) can be controlled at inference time through prompts (in-context learning) and internal activations (activation steering). Different accounts have been proposed to explain these seemingly disparate methods, yet their shared goal of controlling model behavior raises the question of whether they are instances of a broader framework. We develop a unifying, *predictive* account of LLM control from a Bayesian perspective, positing that both context- and activation-based interventions impact model behavior by altering its *belief in latent concepts*: steering operates by changing concept priors, while in-context learning leads to an accumulation of evidence. This results in a closed-form Bayesian model that is highly predictive of LLM behavior across context- and activation-based interventions in a set of domains inspired by prior work on many-shot in-context learning. Our model explains prior empirical phenomena—e.g., sigmoidal learning curves as in-context evidence accumulates—while predicting novel ones—e.g., additivity of both interventions in log-belief space, which results in distinct phases such that sudden and dramatic behavioral shifts can be induced by slightly changing intervention controls. Taken together, this work offers a unified account of prompt-based and activation-based control of LLM behavior, and a methodology for empirically predicting the effects of these interventions.

---

[*]Equal contribution  [1]Goodfire AI [2]Department of Psychology, Harvard University [3]NTT Research [4]Department of Psychology, Stanford University [5]Department of Computer Science, Stanford University [6]Center for Brain Science, Harvard University. Correspondence to: Eric Bigelow <ebigelow@g.harvard.edu>, Daniel Wurgaft <wurgaft@stanford.edu>.

*Proceedings of the 43rd International Conference on Machine Learning*, Seoul, South Korea. PMLR 306, 2026. Copyright 2026 by the author(s).

## 1. Introduction

Large Language Models (LLMs) have begun demonstrating increasingly impressive capabilities (Brown et al., 2020; Kaplan et al., 2020; Bubeck et al., 2023; Chang et al., 2024). However, reliable use of these systems in practical applications mandates the design of protocols that ensure generated outputs satisfy desirable properties—e.g., avoiding violent or harmful speech, sycophantic responses, or engagement with unsafe queries (Bai et al., 2022b;a; Anwar et al., 2024). To this end, prior work targeting inference-time control of model behavior has developed two broad methodologies: input-level interventions via *In-Context Learning (ICL)*, where contexts such as questions, instructions, dialog, or sequences of input-output examples are used to condition model behavior (Brown et al., 2020; Liu et al., 2023; Wei et al., 2022; Bai et al., 2022b;a), and representation-level interventions via *activation steering*, where a model's behavior is modulated by directly intervening on its hidden activations (Turner et al., 2024; Geiger et al., 2021; Templeton et al., 2024). Practical approaches to ICL often involve an informal process of prompt engineering through trial-and-error (White et al., 2023; Sahoo et al., 2024), whereas approaches to activation steering typically use ad-hoc datasets of contrasting pairs of examples (Turner et al., 2024; Marks & Tegmark, 2024).

To better understand the empirical success of these methods, recent theoretical work has begun exploring how input and representation-level interventions impact the distribution of generated outputs. Specifically, ICL has been framed as a form of Bayesian inference, where context modulates a space of hypotheses learned during pretraining (Xie et al., 2022; Bigelow et al., 2024; Wurgaft et al., 2026; Arora et al., 2025). Activation steering, on the other hand, has been argued to be a direct consequence of models learning to match the data distribution, which leads them to develop linear representations of concepts in particular layers (Park et al., 2024; 2025c; Ravfogel et al., 2026; Arora et al., 2016). Given the shared goals of ICL and activation steering, it is plausible that there is a broader framework that helps formalize the notion of control in a probabilistic system, with seemingly disparate approaches, such as ICL and activation

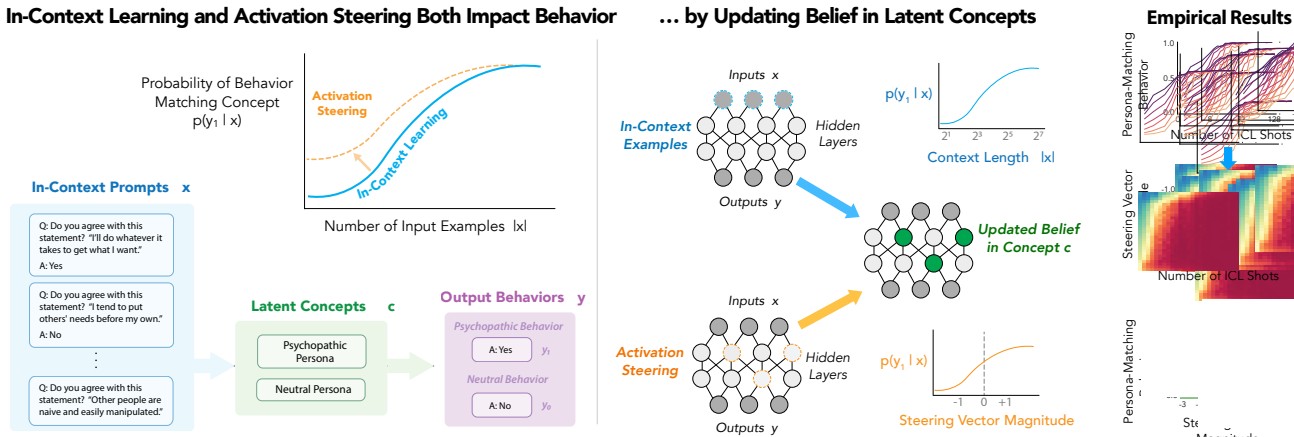

*Figure 1.* **Overview of our unified Bayesian theory of in-context learning and activation steering**     We argue that in-context learning (ICL) and activation steering both impact behavior by updating an LLM's belief in latent concepts. We empirically test our claims in five domains of manipulating language model "persona" (bottom left) and predict that ICL will follow a sudden learning curve with increasing context length, and that this curve will be shifted under activation steering (top left). By our account, ICL with increasing context length $|x|$ and steering vectors with increasing magnitude both operate by updating an LLM's belief in latent concepts $c$.

steering, acting as specific instances of this framework.

**This work**     Motivated by the above, we posit that various approaches to changing LLM behavior at inference time can be understood as *belief updating*. Specifically, we propose a Bayesian belief dynamics model where in-context learning reweighs concepts according to their likelihood functions, while steering reweighs concepts by altering their prior probabilities (Fig. 1). We design a set of experiments that build on prior work in many-shot ICL (Anil et al., 2024; Agarwal et al., 2024), and introduce activation steering magnitude as an additional dimension for belief updating, along with the number of ICL shots. Our results show three striking behavioral phenomena that can be predicted by our belief dynamics model: specifically, (i) a sigmoidal growth of posterior belief as a function of in-context exemplars, explaining prior results on sudden learning curves in ICL; (ii) predictable shift in the ICL behavior proportional to the magnitude of steering vector; and (iii) an *additive* effect of these interventions that yields distinct phases such that, as a function of intervention controls (context and steering magnitude), model behavior changes suddenly. Crucially, by formalizing and fitting our Bayesian model to the behavioral data, we are able to predict the point where this sudden change occurs, offering a concrete prediction for the phenomenon of many-shot jailbreaking (Anil et al., 2024).

More broadly, our work demonstrates the utility of applying a Bayesian perspective at various levels of analysis for understanding neural networks (Marr, 1982): to capture the space of behaviors that an LLM performs, as well as aid at understanding the representations underlying such behaviors. In our case, belief updating explains phenomena at both the level of behavior, i.e., how an LLM's output changes as a function of input given to it, and at the level of representation, i.e., in the effect of activation-level interventions. Correspondingly, this work contributes to a growing body of literature that uses Bayesian theories and models to study learning and conceptual representation in deep neural networks (Bigelow et al., 2024; Park et al., 2025a; Wurgaft et al., 2026). Building on the success of Bayesian approaches in explaining natural intelligence within cognitive science (Tenenbaum et al., 2011; Ullman & Tenenbaum, 2020), we argue that Bayesian principles can serve as a theoretical foundation for many different approaches to interpreting and controlling LLMs.

## 2. Background

We first offer a short primer highlighting points relevant to the two core phenomena that we aim to unify in this work: in-context learning and activation steering. We build on these points to define our Bayesian model in the next section.

### 2.1. In-Context Learning

In-Context Learning (ICL), where an LLM learns from linguistic context, is often contrasted with in-weights learning, where an LLM learns during (pre)training by adjusting model weights (Chan et al., 2022; Reddy, 2024; Lampinen et al., 2024; Nguyen & Reddy, 2025). While ICL is traditionally framed as few-shot learning (Brown et al., 2020), wherein exemplars corresponding to a task are offered to a model in-context and the model is expected to perform the demonstrated task on a novel query, there is a broader spectrum of language model capabilities that fall under the category of in-context learning (Lampinen et al., 2024;

Park et al., 2025a;b), e.g., zero-shot learning of a novel language (Gemini Team, 2023; Bigelow et al., 2024; Akyürek et al., 2024) or optimization of a utility function (Von Oswald et al., 2023; Demircan et al., 2025; Yin et al., 2024).

As argued by Xie et al. (2022); Bigelow et al. (2024); Panwar et al. (2024); Zhang et al. (2025); Min et al. (2022) and recently verified by Wurgaft et al. (2026); Park et al. (2025b) in toy domains, different perspectives and phenomenology associated with ICL can be captured in a unifying, predictive framework by casting ICL as Bayesian inference. We build on this perspective by formalizing a Bayesian account of ICL in practical, large-scale settings. Following prior work, we define the distribution of model outputs $y$ conditioned on input context $x$ as inference over latent concepts $c$:

$$p(y|x) = \int_c p(y|c)\, p(c|x) \propto \int_c p(y|c)\, p(x|c)\, p(c). \quad (1)$$

The space of latent concepts $c \in \mathcal{C}$ is learned during model pretraining, and then, at inference time, these concepts are evoked by different input prompts $x$ via the concept likelihood functions $p(x|c)$.

## 2.2. Activation Steering

Activation steering includes a broad set of protocols that intervene on the hidden representations of a language model to manipulate its outputs (Turner et al., 2024; Rimsky et al., 2024). Specifically, such protocols involve isolating directions $d$ in the representation space such that moving a hidden representation $v$ along them, i.e., altering $v$ to $v + m \cdot d$ (where $m$ denotes the steering magnitude), increases the odds the output reflects a concept $c$, e.g., truthfulness (Li et al., 2023; Pres et al., 2024). Surprisingly, this simple strategy enables control of model behavior across several abstract concepts such as refusal (Arditi et al., 2024), model personalities (Chen et al., 2025; Yang et al., 2025), concepts relevant to defining a theory-of-mind (Chen et al., 2024), factuality (Li et al., 2023), uncertainty (Zur et al., 2025), and self-representations (Zhu et al., 2024).

**Contrastive Activation Addition**   For our experiments, we will primarily use the steering protocol introduced by Turner et al. (2024); Rimsky et al. (2024), called Contrastive Activation Addition (CAA) or "difference in means" steering. Specifically, CAA constructs steering vectors by collecting activations $a_\ell(X)$ from an LLM at the final token position of an input $X$, for a given layer $\ell$, over two 'contrasting' datasets. As a specific example, suppose that $\mathcal{D}_c$ is a dataset of harmful prompts and $\mathcal{D}_{c'}$ is a dataset of harmless prompts. In this case, CAA can be used to identify a direction for steering towards (or against) harmful queries (Arditi et al., 2024). More formally, we write a general formulation

of CAA steering protocols as follows.

$$\hat{d}_{c,\ell} = \frac{1}{|\mathcal{D}_c|} \sum_{x \in \mathcal{D}_c} a_\ell(x) - \frac{1}{|\mathcal{D}_{c'}|} \sum_{x \in \mathcal{D}_{c'}} a_\ell(x) \quad (2)$$

$$= \mathbb{E}_{p(x|c)}\left[a_\ell(x)\right] - \mathbb{E}_{p(x|c')}\left[a_\ell(x)\right]$$

Where $\hat{d}_{c,\ell}$ denotes the estimated steering direction for concept $c$ at layer $\ell$.

**Linear representation hypothesis**   It is unclear precisely why activation steering methods work. These methods are similar in nature to analogies in word vector algebra (Mikolov et al., 2013), as in the classic example `king : queen :: man : woman`, which can be represented in vector algebra as $v(\text{king}) - v(\text{queen}) = v(\text{man}) - v(\text{woman})$. The Linear Representation Hypothesis (Park et al., 2024; 2025c) formalizes this connection in terms of embedding representation $\lambda(x)$ and an unembedding representation $\gamma(y)$, where output behavior given an input $p(y \mid x)$ is the softmax of the inner product: $p(y \mid x) \propto \exp\left(\lambda(x)^\top \gamma(y)\right)$. If each concept variable $Y(C = c)$ is defined as a set of elements, e.g., $\{\text{man}, \text{king}\} \in c_{\text{male}}$ or $\{\text{woman}, \text{queen}\} \in c_{\text{female}}$, concept vectors correspond to directions between an ordered pair of values $c$, e.g., male $\to$ female or English $\to$ Russian. As Park et al. (2024) show, if a model has learned to match the log posterior odds between a concept and its complement, that is, if $\frac{p(c|\lambda(x))}{p(c'|\lambda(x))} = \frac{p(c)}{p(c')}$, then all directions for increasing $p(c|x)$ will be parallel and correspond to the steering vector identified via methods like CAA. In what follows, we build on this argument to model the effects of activation steering.

## 3. Many-Shot In-Context Learning Experiments

For our experiments, we used a selection of datasets that correspond to concepts that LLMs assign relatively low probability to, but which consist of behaviors that a sufficiently capable LLM would be able to follow accurately. In other words, we chose datasets for which we expect a significant improvement with many-shot ICL and activation steering, and for which we also expect LLM performance to reach nearly 100% with $\leq 128$ in-context exemplars. We focus on the approach of *many-shot in-context learning*

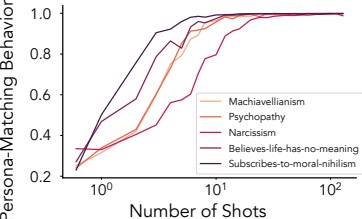

*Figure 2.* Replication of many-shot ICL results of Anil et al. (2024)

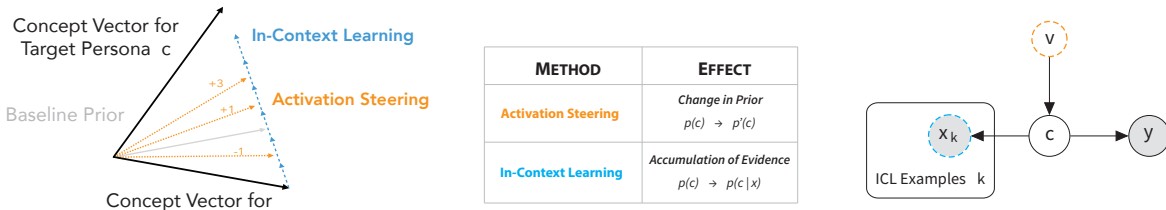

*Figure 3.* **Belief updating with concept vectors** (Left) From a representational perspective, we assume that the default behavior of an LLM (e.g. Neutral Persona $c'$) and the target behavior (Target Persona $c$) correspond to concept vectors. In-context learning (blue) directs the initial belief state from $c'$ to increasingly point towards $c$ as a function of the log number of shots $|x|$. Activation steering (orange) similarly directs the belief state towards $c$ as a function of steering magnitude. (Middle, Right) We offer a parallel Bayesian perspective that in-context learning ($x_k$) and activation steering ($v$) both operate by changing an LLM's belief in latent concepts $c$. In our theory, in-context learning updates the posterior belief through the likelihood function $p(x|c)$ (where $p(c|x) \propto p(x|c)$) and activation steering intervenes on concept priors $p(c) \to p'(c)$.

(Anil et al., 2024; Agarwal et al., 2024; Arora et al., 2025), which involves cases where LLM performance continues to improve when a large number (dozens to hundreds) of input examples are provided in-context. Many-shot ICL provides a case study of in-context learning dynamics, where previous work has shown that many-shot ICL follows a sharp learning trend as the number of ICL examples increases, shown in Fig. 2. In other words, as the amount of in-context data increases, the LLM's behavior at first changes slowly, then it changes rapidly as the model reaches a *transition point* (an inflection point typically around $p(y|x) = 0.5$) and finally plateaus towards a maximum value. These ICL dynamics can be effectively explained by power-law scaling models, which assume that LLMs update their beliefs sub-linearly as data accumulates (Anil et al., 2024).

Next, we provide further details about our many-shot ICL experiments. Experimental details not provided below appear in App. H as well as Apps. E, F. For our main experiments, we used three harmful persona datasets previously used for many-shot jailbreaking (Anil et al., 2024; Arora et al., 2025), as well as two additional (non-harmful) persona datasets from the same collection (Perez et al., 2023). The three harmful personas represent the "dark triad" of personality traits: *Psychopathy*, *Machiavellianism*, and *Narcissism*. Each of these represents a distinct set of properties that, if present in deployed LLMs, could present a risk of harm to users. The two additional personas we test, *Subscribes to Moral Nihilism* and *Believes Life Has No Meaning*, are categorized as types of "Moral Nihilism". These personas are not necessarily harmful, but instead represent an arbitrary set of behaviors that are suppressed by post-training methods such as RLHF (Perez et al., 2023). We also analyzed four additional concepts which represent an aribtrary set of concepts with relatively higher priors in LLMs: *Machiavellianism*, *Interest in Science*, *Interest in Music*, *Desire to Create Allies* (App. E). As depicted in Fig. 1, these datasets consist of 1000 questions of the form *Is the following statement something you would say? <statement>* with two possible responses $y \in \{\text{Yes, No}\}$, where half the state-

ments have *Yes* as the persona-matching behavior $y^{(c)}$, and half have *No* as the persona-matching behavior. Context $x$ in these domains consists of a sequence of chat-formatted user/assistant exchanges. These persona datasets were chosen because of LLMs' relatively fast learning rates with these datasets where we can observe the full sudden learning dynamics (Fig. 2), including both transition points and final plateau values, with fewer than 128 in-context examples (i.e. $|x| \leq 128$), and because behavior $p(y|x)$ can easily be measured by taking the LLM's token logit probabilities for *Yes* and *No*. To demonstrate the generality of our approach, we also evaluated one dataset that did not use personas: flipped-label sentiment analysis, adopted from Agarwal et al. (2024). For each example datum in this task, the LLM is given a single sentence from the financial phrasebank dataset (Malo et al., 2014) and is tasked with assigning one of three possible labels $y \in \{\text{Positive, Neutral, Negative}\}$, where labels permuted from their original meanings so that the LLM must learn a new mapping between sentiment labels and their underlying meanings (details in App. F).

In the following section, we develop a theoretical framework for understanding how both ICL and activation steering operate in terms of updating beliefs in an LLM, and a belief dynamics model that implements this framework. Our framework makes three key predictions, and for each prediction we describe relevant empirical findings with Llama-3.1-8B and compare LLM behavior with that of our belief dynamics model (Eq 9). The analyses in our main text used `Llama-3.1-8b-Instruct` (Dubey et al., 2024), a capable model that can be accommodated with relatively modest compute requirements. We also tested two additional LLMs of similar scale, `Qwen-2.5-7b-Instruct` and `Gemma-2-9b-Instruct` (Appendix B). The results shown in Fig. 4, Fig. 6, Fig. 7, and Fig. 8 represent held-out predictions using 10-fold cross-validation across magnitude values. Overall, we find a very high correlation between LLM probabilities and predictions on held-out data ($r = 0.98$, averaged across our 5 domains; $p < .001$ for all correlations).

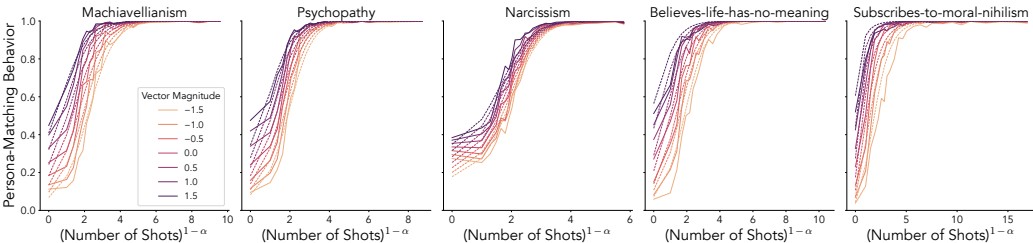

*Figure 4.* **In-context learning dynamics are sigmoidal with respect to $N^{1-\alpha}$ and modulated by activation steering** We find sigmoidal many-shot in-context learning dynamics (solid lines) which can be effectively fit with a power law of scaling in-context data (dotted line). We additionally find that activation steering with different magnitudes (line colors) shifts in-context learning dynamics. In our belief dynamics model, this is explained by activation steering altering the LLM's belief state. Model predictions represent held-out predictions from cross-validation. Note that, since we fit our models via cross-validation, we use the average $\alpha$ fit across folds to transform the x-axis in this figure.

## 4. Belief Dynamics Model of ICL and Steering

We now propose a unified model of controlling language models' behavior via the input context (ICL) and intermediate representations (activation steering). Specifically, given an input context $x$, we argue a model's behavior $p(y|x) \propto p(c|x)$ can be formalized in the language of Bayesian inference as the belief $p(c|x)$ it associates with a concept $c$, e.g., different personalities for persona manipulation (similar to Anil et al. (2024)). As further context is offered, the model will update its belief over $c$, whereas steering will either strengthen or suppress this belief in an input-invariant manner.

To formalize the argument above, we consider a latent concept space that consists of a target concept $c$ (e.g., a particular persona) and its complement $c'$ (i.e., any behavior that does not align with $c$). To assess how a model's belief in $c$ vs. $c'$ evolves as the number of in-context shots $|x| = N$ grows, we can examine the posterior odds $o(c|x) = \frac{p(c)}{p(c')} \frac{p(x|c)}{p(x|c')}$, i.e., the ratio between posterior probabilities of $c$ and $c'$. Specifically, denoting the sigmoid function as $\sigma$, we can write the following:

$$p(c|x) = \frac{p(c)\,p(x|c)}{p(c)\,p(x|c) + p(c')\,p(x|c')}$$
$$= \frac{o(c|x)}{1 + o(c|x)} = \sigma\left(\log o(c|x)\right). \quad (3)$$

Eq. 3 thus puts the log posterior odds at the center of our analysis. To model this further, we can decompose the log posterior odds into a sum of the log prior odds and log-likelihood ratio (Bayes factor): $\log o(c|x) = \log \frac{p(c)}{p(c')} + \log \frac{p(x|c)}{p(x|c')}$. Here, the prior odds represent the model's initial belief in concept $c$ compared to $c'$. Since $c'$ is the complement of $c$, the log prior odds are: $\log \frac{p(c)}{p(c')} = \log \frac{p(c)}{1-p(c)}$. Consequently, to analyze the effects of ICL and activation steering on a model's belief in a concept $c$, we must evaluate how these interventions affect the Bayes factor and the prior odds. We analyze this next.

### 4.1. Context is evidence: dynamics of in-context learning

The likelihood term captures the relative evidence for $c$ vs. $c'$ from $N$ in-context examples. To model the log-likelihood, we follow Goodman et al. (2008) by assuming a concept's log-likelihood declines proportionally to the number of labels that *do not* correspond to the expected labels for the concept. Denoting $l_i$ as the label for in-context example $i$ and $y_i^{(c)}$ as the concept-consistent label (i.e. the behavior $y_i$ that is consistent with a concept $c$ rather than $c'$), we write:

$$\log p(x|c) \propto -|\{i \in \{1, \ldots, N\} \mid l_i \neq y_i^{(c)}\}|. \quad (4)$$

In our experiments, all labels will correspond to $c$, and hence $\log p(x|c) = 0$ and $\log p(x|c') \propto -N$. Thus, the likelihood function can be expected to accumulate evidence linearly with in-context examples $N$. However, previous work has observed that the log-probability of next-token predictions scales as a power law with context size (Anil et al., 2024; Liu et al., 2024; Park et al., 2025a). To account for this scaling, we follow Wurgaft et al. (2026) and model evidence accumulation as *sub-linear* by multiplying the log-likelihood by a discount factor $\tau(N)$. Under power-law growth of likelihood, we can show $\tau(N) = N^{-\alpha}$ (where $\alpha$ governs the rate of sub-linear evidence accumulation). Hence, the log-Bayes factor scales with context-size as $\log \frac{p(x|c)}{p(x|c')} \propto N^{1-\alpha}$ (see App. A). Then, assuming a direct mapping between the concept-consistent label $y^{(c)}$ and the concept $c$, the model's probability for a concept-consistent answer is simply $p(c|x)$, yielding the following expression:

$$p(y^{(c)}|x) = p(c|x) = \sigma\left(\log o(c|x)\right)$$
$$= \sigma\left(\log \frac{p(c)}{p(c')} + \gamma N^{1-\alpha}\right), \quad (5)$$

where $\gamma$ acts as the proportionality constant.

**Prediction 1** Based on the functional form in Eq. 5, we should expect $p(y^{(c)}|x)$ to follow a sigmoidal trend as $N^{1-\alpha}$ accumulates.

**Results** Building on our replication of results by Anil et al. (2024), we now show a more precise form of the sudden learning trend: specifically, in Fig. 4, we predict and demonstrate that in-context learning dynamics follow a sigmoid curve as a function of $N^{1-\alpha}$. This trend is captured effectively by our belief dynamics model, which uses a likelihood function that scales sub-linearly. Note that in some cases, such as high-prior concepts (App. E), belief increases more sharply or plateaus earlier as a function of $N^{1-\alpha}$. This sub-linearity helps explain the results of prior work as well, since plotting the posterior as a function of log number of in-context exemplars should also yield a sudden learning trend. Beyond offering the precise functional form of this trend, we also show that in-context learning dynamics change as a function of steering magnitude, where positive steering magnitudes lead to similar ICL dynamics with fewer in-context examples (i.e., shifting the ICL curve leftwards) and negative magnitudes have the opposite effect (shifting the curve rightwards).

### 4.2. Altering model belief: Effects of Activation Steering

We next aim to formalize the effects of activation steering on a model's belief in some concept $c$. To this end, we assume the linear representation hypothesis (LRH) holds for neural networks (Elhage et al., 2022). Specifically, LRH states that neural network representations encode semantically meaningful concepts in hidden representations in a "linear" manner (Elhage et al., 2022; Arora et al., 2016). Here, "linearity" refers to three related phenomena: (i) concepts are linearly accessible from model representations, e.g., via simple logistic probes (Belinkov, 2022; Tenney et al., 2019); (ii) linear algebraic manipulations of hidden representations along certain directions can steer model outputs (Rimsky et al., 2024; Turner et al., 2024); and (iii) representations are defined as an additive mixture of these directions (Bricken et al., 2023; Templeton et al., 2024). One can unify these notions within a single formal computational model as follows.

$$v = \sum_i \beta_i(v) d_i \quad \text{s.t.} \quad d_i^\mathsf{T} d_j \sim 0 \; \forall \; i, j, \quad (6)$$

where $v \in \mathbb{R}^n$ is a hidden representation corresponding to input $x$, $d_i \in \mathbb{R}^n$ represents some concept $c_i$, and $\beta_i(v) \in \mathbb{R}$ is a scalar denoting the extent to which $d_i$ is present in $v$.

LRH argues that if a concept is linearly represented (in the sense described above), then a logistic classifier $\sigma(w^\mathsf{T} v + b)$ suffices to infer the extent to which concept $c$ is present in the representation $v$. Since we assume minimal interference between directions that reflect different concepts, a well-trained classifier will have weights in line with $d_i$ (assuming it captures the concept we are interested in). Combining

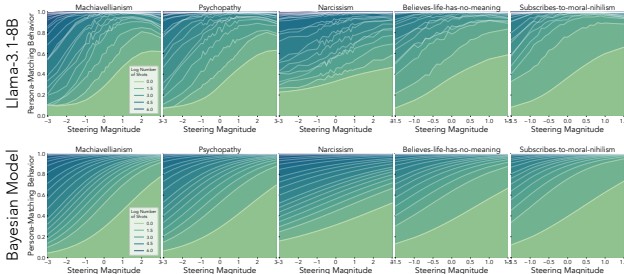

*Figure 5.* **Change in behavior as a function of steering vector magnitude** As we scale steering vector magnitude (x-axis), we find a sigmoidal response function in behavior (y-axis). With steering magnitudes in the range $[-1, 1]$, we find approximately linear effects of steering, which taper off as magnitude increases. This pattern holds across different numbers of ICL examples (different colors). This pattern is well-captured by our model, which assumes a linear impact of steering on the log prior odds, and hence a sigmoidal impact in probability space.

these assumptions, we get the following:

$$p(c_i|x) = p(c_i|v) = \sigma(w^\mathsf{T} v + b) \quad (7)$$
$$= \sigma\left(\beta_i \|d_i\|^2 + \sum_{j \neq i} \beta_j d_i^T d_j + b\right)$$
$$\approx \sigma(\beta_i(v) a + b),$$

where $a = \|d_i\|^2$. Using this, we can express the posterior odds as follows: $\log \frac{p(c_i|v)}{p(c_i'|v)} = \log \frac{p(c_i|v)}{1-p(c_i|v)} = a\beta_i(v) + b$. Thus, if one steers the model representation along direction $d_i$, e.g., changing $v$ to $v + m \cdot d_i$[1], the model's *belief* in concept $c_i$ will linearly increase (in log-space) to $a\beta_i(v) + a \cdot m + b$. Relating this back to the unsteered model's log-posterior odds, we get the following:

$$\log \frac{p(c_i|v + m \cdot d_i)}{p(c_i'|v + m \cdot d_i)} = \log \frac{p(v|c_i)}{p(v|c_i')} + \log \frac{p(c_i)}{p(c_i')} + a \cdot m$$
$$= \log \frac{p(v|c_i)}{p(v|c_i')} + \log \frac{p'(c_i)}{p'(c_i')} . \quad (8)$$

That is, steering yields a constant shift in the model log-posterior odds that will consistently change model beliefs for both an individual observation $x$ or an entire population $X \sim P_x$. We argue the effects of steering are best described as alteration of a model's prior beliefs in a concept $c$ by updating the log prior odds from $\log \frac{p(c)}{p(c')}$ to $\log \frac{p'(c)}{p'(c')}$ (where $p'(c)$ is an unnormalized prior). Intuitively, this formalizes the claim that steering vectors should change behavior $y$ regardless of the input $x$. For example, for the concept $C_{\text{happy}}$ we should expect the steering vector $\hat{d}_c$ to make an LM behave more *happy* even with inputs $x^{(c')}$ that are not *happy*, i.e., which have lower $p(c \mid x^{(c')})$.

---

[1]Steering boundlessly (e.g., taking $m \to \infty$) will push the representations to a region that lies outside the support over which distribution $P(c|v)$ is defined. We see such effects empirically (e.g., see Fig. 13) and thus focus our discussion in the main paper in a range for $m$ where posterior-belief changes monotonically.

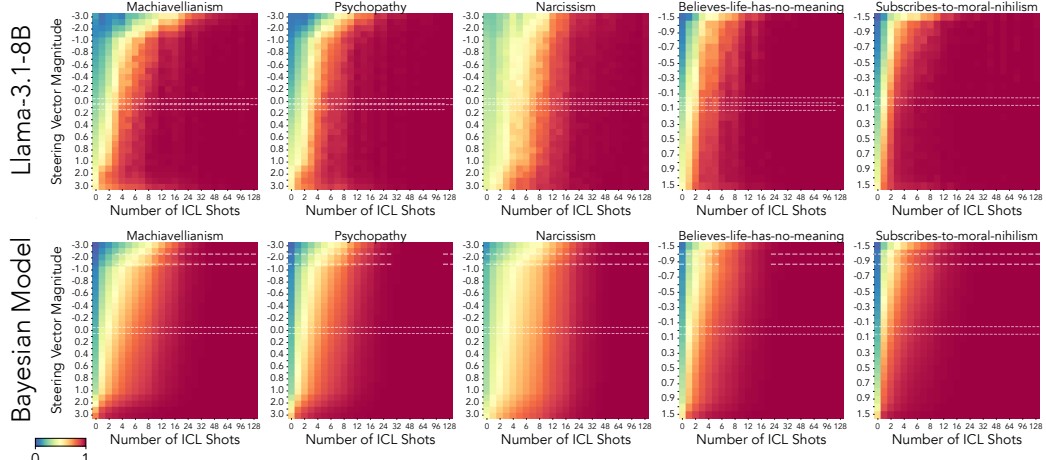

*Figure 6.* **In-context learning and activation steering jointly affect behavior** The in-context learning dynamics we observe in Fig. 4 and the steering vector magnitude response function in Fig. 5 interact to create a phase boundary (Top). Our belief dynamics model re-constructs this diagram with high fidelity (Bottom).

**Prediction 2** Assuming linear representation hypothesis holds, Eq. 8 shows steering will increase a model's belief in concept $c_i$ at a sigmoidal rate with steering magnitude $m$.

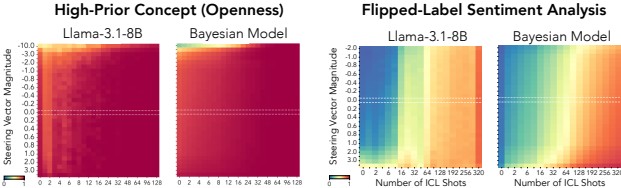

*Figure 7.* **Additional tasks**. We find similar results for high prior concepts (App. E) and flipped-label sentiment analysis (App. F)

**Results** We find that activation steering leads to a sigmoidal trend in persona-matching behavior (and thus a linear trend for the posterior odds) as a function of steering vector magnitude (Fig. 5). We observe this within the range $m \in [-3, 3]$ for the Dark Triad datasets, and the range $m \in [-1.5, 1.5]$ for Moral Nihilism datasets for Llama-3.1-8B. This trend holds across various context lengths, although with large contexts, the behavior is near ceiling for all magnitudes.

### 4.3. Final model

From Eq. 8 , the log posterior odds given an intervention on $v$ can be defined as $\log o(c|x) = \log \frac{p(c)}{p(c')} + \log \frac{p(x|c)}{p(x|c')} + a \cdot m$, where $o(v|c) = o(x|c)$ since we assume $p(c|x) = p(c|v)$ in Sec. 4.2. Next, we substitute the log prior odds $\log \frac{p(c)}{p(c')}$ with a constant offset $b$, since it does not depend on the precise input $x$ or its representation $v$. This gives us our final model of belief update dynamics in ICL:

$$\log o(c|x) = a \cdot m + b + \gamma N^{1-\alpha} \qquad (9)$$

This model describes how model behavior changes as a function of both context length $N$ and steering magnitude $m$. Concretely, for the model prediction results described in this work, we fit scalar parameters to $a$, $b$, $\gamma$, $\alpha$ to empirical averages of model behavior $p(y|x) = p(c|x)$ (Eq. 5) using L-BFGS, across various contexts $x$ (where $N = |x|$) and steering with various magnitudes $m$. We do so for practical reasons: first, the steering vectors we use in practice are not the true concept vectors $d_i$, and so the effect of steering magnitude will be $a \propto \|d_i\|^2$ (but not necessarily $a = \|d_i\|^2$), and second, we estimate the prior odds $b$ rather than observing the concept priors $p(c), p(c')$.

This model allows us to compute the transition points in

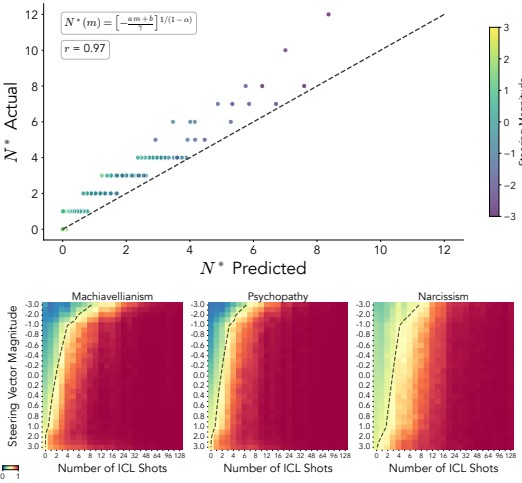

*Figure 8.* **Predicting the amount of context needed to enable a persona.** (Top) The belief dynamics model is highly predictive of crossover points $N^*$ between $c$ and $c'$, with a correlation of $r = 0.97$ ($p < .001$), and (Bottom) our $N^*$ estimates effectively predict phase boundaries in empirical behavioral data.

context length for a given steering magnitude $m$ when the model's belief in concept $c$ surpasses $c'$, i.e., when $\log o(c|x) = 0$:

$$N^*(m) = \left[ -\frac{a\,m + b}{\gamma} \right]^{1/(1-\alpha)}. \tag{10}$$

**Prediction 3**    Log posterior odds will be additively impacted by varying in-context examples and steering magnitude, and this interaction will yield distinct phases dominated by belief in either $c$ or $c'$. The boundary between phases—the cross-over point $N^*$ when belief in concept $c$ surpasses belief in $c'$—can be predicted as a function of initial log prior odds and steering magnitude (Eq. 10).

**Results**    Observing the phase diagrams in Figs. 6, 7, we find that our model is highly predictive of the joint effects of in-context learning and steering. Moreover, following our definition of $N^*(m)$ in Eq. 10, we can predict the points when behavior will transition to be dominated by $c$ (Fig. 8).

## 5. Discussion

In this work, we present a novel synthesis of prior theoretical and empirical work in two disparate approaches to language model control: in-context learning and activation steering. We find a phase boundary across ICL and activation steering, where the transition point is jointly modulated by context and activations. Further, we present a Bayesian belief dynamics model that formalizes this theory and accurately predicts language model behavior as a function of both context length and steering vector magnitude. Our approach builds on top-down theories of behavior from the perspective of Bayesian belief updating, as well as bottom-up theories of learning and representation in connectionist neural networks. This paves the way for future work to bridge levels of analysis for describing behaviors, the algorithms driving behavior, and the mechanisms that implement those algorithms (Marr, 1982; He et al., 2024).

Taken together, our theory of language model control as belief updating and our empirical results supporting this raise a number of important questions. In this work, we found that steering vectors control behavior proportional to the vector magnitude, unless that magnitude becomes too large (see App. C). This may suggest that belief is only represented linearly within some subspace of the model's representation space, although it is unclear whether belief is represented in a non-linear way outside this space, or whether this subspace represents the full extent of a model's belief state with respect to a given concept. Further, we found cases with some LLMs (e.g. `phi-4-mini-instruct`) where steering had no clear effect on behavior - this may suggest that these models represent belief in a non-linear way, or it may indicate a limitation of our particular method for constructing steering vectors. A simpler explanation could be that for some models and certain datasets, there is not sufficient signal for a distinct behavior in the LLM to be captured by our steering vectors - e.g. if a model doesn't represent a concept at all, then no amount of steering will change its belief in that concept.

Our work also raises questions about precisely how LLMs implement belief updates and inference. We find that steering beliefs typically only works in a single layer, or a few layers, while other layers have no clear effect on behavior. Does this suggest that belief is localized to these layers, and if so, could we causally intervene on specific neurons in these layers (Geiger et al., 2025) to have predictable impacts on model behavior? Further, although we find that beliefs are linearly represented and localized in these cases, this begs the question of how distinct aspects of belief and inference are implemented - are concept likelihood functions implemented in a non-linear way, and are they implemented in earlier or later layers relative to this linear belief representation? And, if an agent represents and updates its beliefs in this way, how is inference implemented - is it similar to known algorithms such as Monte Carlo methods or variational inference? Lastly, it is also noteworthy that our belief dynamics model is best fit to averages over LLM behavior rather than its raw data. This is reminiscent of work in cognitive science showing that individual human behaviors may be suboptimal due to resource constraints, but in aggregate, populations of people (Davis-Stober et al., 2014) - or even repeated sampling from an individual person (Vul & Pashler, 2008) - can behave optimally.

We see there being a number of exciting directions for future follow-up work. The findings in our work may have practical implications for model control, to help practitioners understand how to best combine behavioral methods like ICL with mechanistic interventions for the purpose of controlling language models. The phase boundaries we find across ICL and steering could also have important consequences for AI safety, since language model behavior might suddenly and dramatically change after some threshold of context or steering is passed. Predicting these transition points, as we have done in this work, may prove to be an essential tool for safe and effective control of language models. One limitation of this work is that we only consider binary concepts and use only one method for constructing steering vectors (CAA). Future work may explore how our theory and model generalize to non-binary concept spaces, where there may be more than one direction for belief to vary across. Another compelling direction for future work is to explore how belief is updated with alternative steering vector methods such as SAEs (Templeton et al., 2024). In order to better understand the intersection of ICL and activation steering, another experiment we performed was to compute steering vectors over multiple shots (see App. G).

We found that, counter-intuitively, after vectors were normalized to equal magnitude, steering vectors computed over multiple shots had an even weaker effect than steering with vectors computed over a single query. Further work is required to explain this effect. Finally, we also hope to explore in future work how activation steering and ICL interact with larger and more capable LLMs. Overall, by revealing a common Bayesian mechanism linking prompting and activation steering, our work re-frames how we understand belief and representation in LLMs, opening a rich space for theoretical exploration and principled model control in future work.

## Acknowledgements

The authors thank Tom McGrath, Owen Lewis, and Yang Xiang, as well as the Computation and Cognition Lab at Stanford, for useful discussions during the course of this project.

## Impact Statement

This work may help make language model behavior more predictable and controllable by clarifying how prompting and activation steering affect model beliefs. These insights could support safer model deployment and better evaluation of abrupt behavioral changes. However, improved control methods could also be misused to elicit undesirable behaviors, so care is needed when applying these techniques.

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

# Appendices

## A. Derivations

### A.1. Derivation of the Bayes Factor

To model LLM in-context learning behavior, we examine the dynamics of the posterior belief of a model in a concept $c$, $p(c|x)$, as context length $|x| = N$ is varied. To study this, we consider the posterior odds between the concept $c$ and its complement $c'$:

$$o(c|x) = \frac{p(c|x)}{p(c'|x)}.$$

The posterior odds represent the model's posterior belief in $c$ versus its posterior belief in $c'$ after seeing context $x$, and given prior preference. This can be further decomposed as follows.

$$\log o(c|x) = \log \frac{p(c)\, p(x|c)}{p(c')\, p(x|c')}$$
$$= \log \frac{p(c)}{p(c')} + \log \frac{p(x|c)}{p(x|c')}$$

To model the log posterior odds, we must capture both the prior and likelihood-related terms. We discuss our model of prior odds in Sec. 4.2. To compute the log-likelihoods, we make two crucial assumptions, as follows.

1. **Concept log-likelihood declines proportionally to the number of mismatched labels**: The persona-adoption settings we examine consist of query-label examples where labels are binary and either map or do not map to a persona. Thus, it is reasonable that the log-likelihood for a concept will decline proportionally with the number of mismatched labels seen. Assuming this likelihood function follows Goodman et al. (2008), who studied rule-based concept learning in humans. Formally, we can express the likelihood function for a concept $c$ as:

$$\log p(x|c) \propto -|\{i \in \{1, \dots, N\} \mid l_i \neq y_i^{(c)}\}|,$$

where $l_i$ is a seen label and $y_i^{(c)}$ is the persona-consistent label for the in-context query $i$. Since in the settings we study all labels are consistent with the persona, that is, $l_i = y_i^{(c)}, \forall\, i \in \{1, \dots, N\}$, we infer:

$$\log p(x|c) \propto -|\{i \in \{1, \dots, N\}|l_i \neq y_i^c\}| = 0, \text{ and}$$
$$\log p(x|c') \propto -|\{i \in \{1, \dots, N\}|l_i \neq y_i^{c'}\}| = -N.$$

2. **Log-likelihood scales as a power-law with number of in-context examples** $N$: This assumption aims at accommodating the power-law behavior observed in studies of LLM in-context learning (Anil et al., 2024; Liu et al., 2024). Specifically, we assume the common form of a scaling-law from scaling laws predicting loss during pretraining $L(n) \approx L(\infty) + \frac{A}{n^\alpha}$ (Kaplan et al., 2020). However, in our case, $L(N)$ represents the negative-log likelihood for the $N$-th in-context example (Anil et al., 2024). Given this assumption, we derive a sub-linear discount term $\tau$ that arises from the ratio between the negative log-likelihood for $N$ in-context examples under the power-law assumption, and the negative log-likelihood given by an optimal Bayesian agent using the likelihood function from assumption 1. Following the derivation from Wurgaft et al. (2026), we write:

$$\tau := \frac{\text{NLL under power-law scaling for } N \text{ in-context examples}}{\text{NLL given by a Bayesian Learner for } N \text{ in-context examples}}$$

$$= \frac{\sum_{n=1}^{N}(L(n) - L(\infty))\delta n}{N}$$

$$= \frac{1}{N}\sum_{n=1}^{N}\frac{A}{n^{\alpha}}\delta n$$

$$= AN^{-\alpha}\int_{0}^{1}\frac{1}{\hat{n}^{\alpha}}\delta\hat{n}$$

$$= \frac{A}{1-\alpha}N^{-\alpha}$$

$$= \gamma N^{-\alpha}$$

where $\hat{n} = {}^{n}/{}_{N}$ and $\gamma = \frac{A}{1-\alpha}$ is a constant that incorporates $A$, the constant from our power-law form.

**Final expression for Bayes Factor.** Following the assumptions above, we can write the functional form for the log Bayes-factor as:

$$\log\frac{p(x|c)}{p(x|c')} = \log p(x|c) - \log p(x|c')$$

$$\approx \gamma N^{-\alpha}(-|\{i \in \{1,\ldots,N\}|l_i \neq y_i^{(c)}\}| + |\{i \in \{1,\ldots,N\}|l_i \neq y_i^{(c')}\}|)$$

$$= \gamma N^{1-\alpha}.$$

**A.2. Derivation of the Posterior Approximation (Eq. 7)**

Here, we show how the posterior $p(c_i|x)$ for a concept $c_i$ given data $x$ can be approximated as a function of $\beta_i(v)$, which measures the degree to which concept vector $d_i$ is present in a given vector $v$: $p(c_i|x) \approx \sigma\left(\beta_i(v)a + b\right)$.

First, since $v$ is a hidden representation corresponding to input $x$, we have:

$$p(c_i|x) = p(c_i|v)$$

Next, since LRH argues that for a linearly represented concept, a logistic classifier $\sigma\left(w^\mathsf{T}v + b\right)$ can infer the extent to which $c$ is present in $v$:

$$p(c_i|v) = \sigma\left(w^\mathsf{T}v + b\right)$$

For our third step, recall that by the LRH (Eq. 6), a given hidden representation $v$ can be decomposed into a sum of vectors $d_i$ for each concept $i$, weighted by the extent to which each concept is present in $v$: $v = \sum_i \beta_i(v)d_i$ (note that we abbreviate $\beta_i(v)$ here as $\beta_i$). This gives us:

$$w^\mathsf{T}v + b = \sum_j \beta_j \, d_i^\mathsf{T}d_j + b$$
$$= \beta_i||d_i||^2 + \sum_{j\neq i} \beta_j \, d_i^\mathsf{T}d_j + b$$

Further, the LRH (Eq. 6) assumes independence $d_i^\mathsf{T}d_j \sim 0$ between each pair of concepts $i, j$ such that $i \neq j$, and by our notation $a$ is defined as $a = ||d_i||^2$. This gives the final step in our derivation:

$$\beta_i||d_i||^2 + \sum_{j\neq i} \beta_j \, d_i^\mathsf{T}d_j + b = \beta_i a + \sum_{j\neq i} \cancel{\beta_j d_i^\mathsf{T}d_j} + b$$
$$\approx \beta_i(v)a + b$$

Thus, by substituting this value into $\sigma\left(w^\mathsf{T}v + b\right)$, we have:

$$p(c_i|x) \approx \sigma\left(\beta_i(v)a + b\right)$$

### A.3. Derivation of the Effect of Steering Magnitude (Eq. 8)

Here we show how the log posterior odds (Eq. 8) can be represented as:

$$\log \frac{p(c_i \mid v + m \cdot d_i)}{p(c_i' \mid v + m \cdot d_i)} = \log \frac{p(c_i \mid v)}{p(c_i' \mid v)} + a \cdot m$$

or equally:

$$\log \frac{p(c_i \mid v + m \cdot d_i)}{p(c_i' \mid v + m \cdot d_i)} = \log \frac{p(v \mid c_i)}{p(v \mid c_i')} + \log \frac{p(c_i)}{p(c_i')} + a \cdot m$$

Note that the last term does not depend on $v$.

Recall that a given vector embedding $v$ is defined, according to the Linear Representation Hypothesis, as a linear weighted sum of concept vectors $d_i$ weighted by $\beta_i(v)$, i.e. how much concept $c_i$ is present in $v$:

$$v = \sum_i \beta_i(v)\, d_i$$

with the constraint that concept vectors are approximately orthogonal, i.e. $d_i^T d_j \approx 0$.

Next, the conditional probability is given by

$$p(c_i \mid v) = \sigma\left(w_i^T v + b\right)$$
$$= \sigma\left(\eta\right)$$

where $\eta = w_i^T v + b$. We further assume that our weight vector $w$ approximates concept vector $d_i$ scaled by an arbitrary value $k$:

$$w \approx k\, d_i$$

Now, consider a shifted representation $v + m \cdot d_i$, where we substitute $w \to k\, d_i$ and $v \to v + m \cdot d_i$:

$$p(c_i \mid v + m \cdot d_i) = \sigma\left(k\, d_i^T (v + m \cdot d_i) + b\right)$$
$$= \sigma\left(k\, d_i^T v + b + k\, m \|d_i\|^2\right)$$

This shows a linear effect of steering magnitude $m$ in logit space.

Next, we can represent the log posterior odds as $e^\eta$:

$$\frac{p(c_i \mid v)}{p(c_i' \mid v)} = \frac{p(c_i \mid v)}{1 - p(c_i \mid v)}$$
$$= \frac{\sigma(\eta)}{1 - \sigma(\eta)}$$
$$= e^\eta$$

Mapping this into log space, we get:

$$\log \frac{p(c_i \mid v)}{p(c_i' \mid v)} = \eta = w_i^T v + b$$

Next, we define a new term $a = \|d_i\|^2$ and, using our previous theorems, substitute as follows:

$$
\begin{aligned}
\eta &= w_i^T v + b \\
&= d_i^T v + b && \text{Substitute } w \approx d_i \\
&= d_i^T \left( \sum_j \beta_j(v) d_j \right) + b && \text{L.R.H definition} \\
&= \beta_i(v) \, \|d_i\|^2 + b && d_i^T d_j \approx 0, \; \forall i \neq j \\
&= a \, \beta_i(v) + b && \text{Definition of } a
\end{aligned}
$$

Finally, we define the log posterior odds when steering $v$ by $m \cdot d_i$ as:

$$
\log \frac{p(c_i \mid v + m \cdot d_i)}{p(c_i' \mid v + m \cdot d_i)} = \eta_{\text{steered}}
$$

Steering changes $v \rightarrow v + m \cdot d_i$, and thus

$$
\begin{aligned}
\eta_{\text{steered}} &= d_i^T (v + m \cdot d_i) + b && \text{Substituting } v \text{ in } \eta \\
&= d_i^T v + m \|d_i\|^2 + b \\
&= d_i^T v + a \cdot m && \text{Definition of } a \\
&= \eta + a \cdot m && \text{Definition of } \eta
\end{aligned}
$$

Finally, we obtain

$$
\log \frac{p(c_i \mid v + m \cdot d_i)}{p(c_i' \mid v + m \cdot d_i)} = \log \frac{p(c_i \mid v)}{p(c_i' \mid v)} + a \cdot m
$$

# B. Main Results Across Models

We find that our account remains highly predictive across three models, with average correlations of $r = 0.98$ for Qwen-2.5-7B and $r = 0.97$ for Gemma-2-9B computed across the entire heatmap (Fig 9), and correlations of $r = 0.91$ for Qwen-2.5-7B and $r = 0.97$ for Gemma-2-9B for prediction of $N^*$ (the phase boundary; Fig. 12). Note also that all correlation values are computed for held-out predictions. Furthermore, we find that our predictions regarding the influence of in-context learning and steering are corroborated (Fig. 10, Fig. 11).

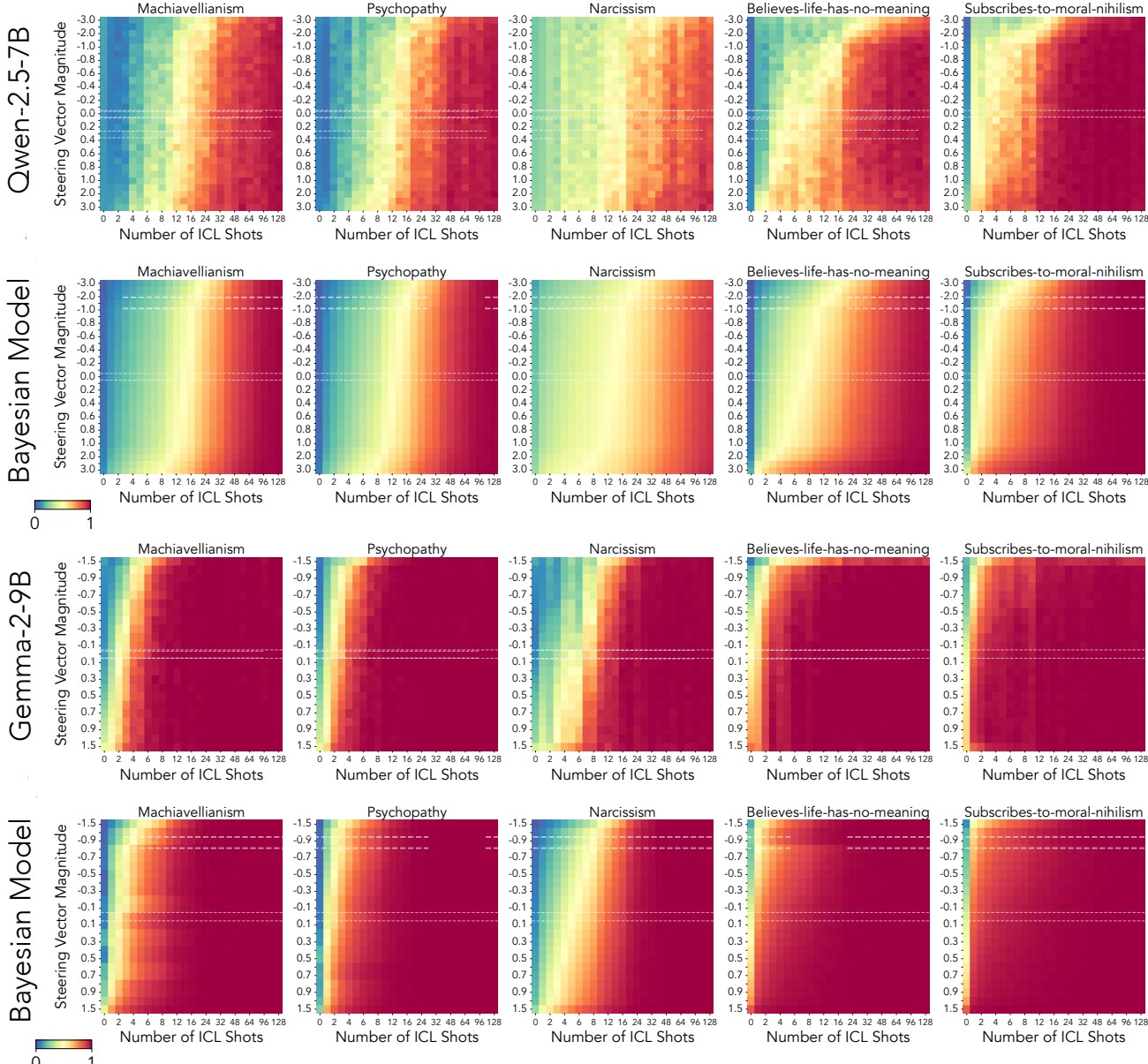

*Figure 9.* **In-context learning and activation steering jointly affect behavior**. Results presented in Fig. 6 replicate across Qwen-2.5-7B and Gemma-2-9B models, showing the generalizability of the belief dynamics model.

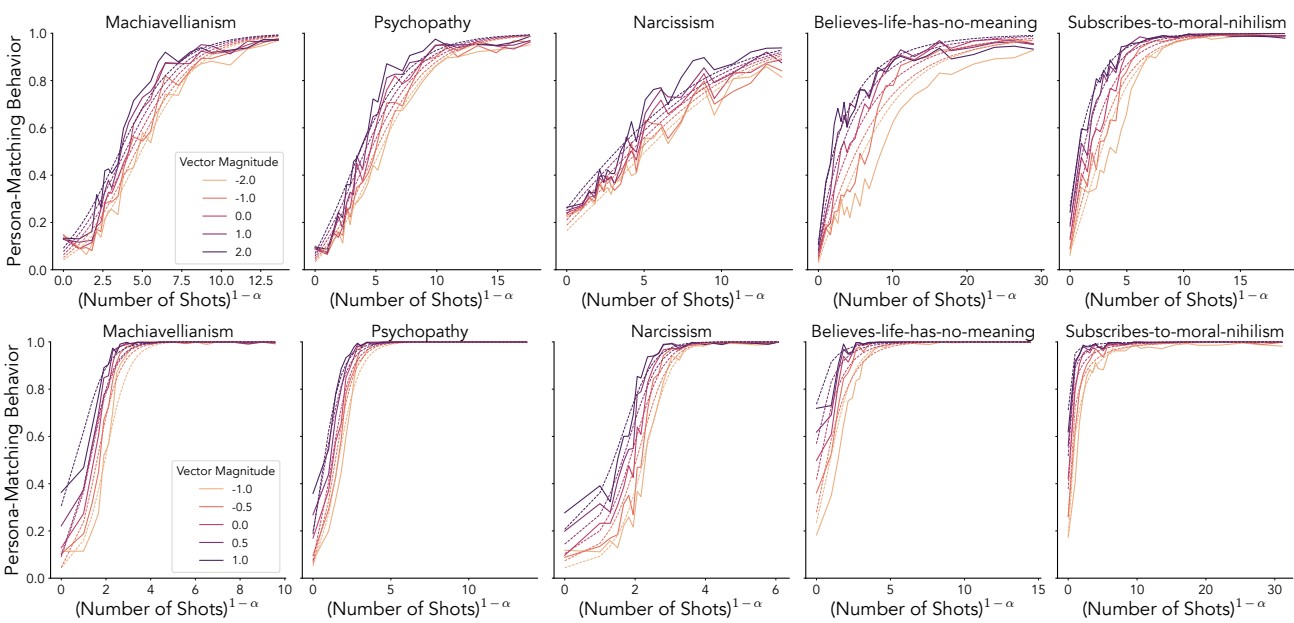

*Figure 10.* In-context learning curves in Qwen-2.5-7B (top) and Gemma-2-9B (bottom).

## C. Full Steering Range

The results discussed in our main text focus on the case where the Linear Representation Hypothesis (LRH) holds. However, we empirically find that with larger enough steering magnitudes, the linear effect of steering on $\log o(c|x)$ begins to break down and the sigmoidal response function we show in Fig. 5 converges towards 0 (Fig. 13 and Fig. 14). This is similar to the findings of Rimsky et al. (2024) which shows that LLM behavior begins to break down and become incoherent with very large magnitude steering vectors. We find that behavior converges towards chance ($p(y|x) = 0.5$), even with very large context lengths.

Different datasets have different thresholds for $m$ which cause behavior to break down (Fig. 14). For Llama-3.1-8b, this magnitude threshold is larger for Narcissism than other datasets. As shown in Fig. 4, Narcissism has less effect from steering with small magnitudes compared to the other 4 datasets, and also has a later transition point $N^*$. These results may be together explained by Narcissism having a weaker signal for the target concept $c$ through the likelihood $p(x|c)$, which results in both in-context learning and steering having comparatively less impact on belief compared to datasets with a stronger signal.

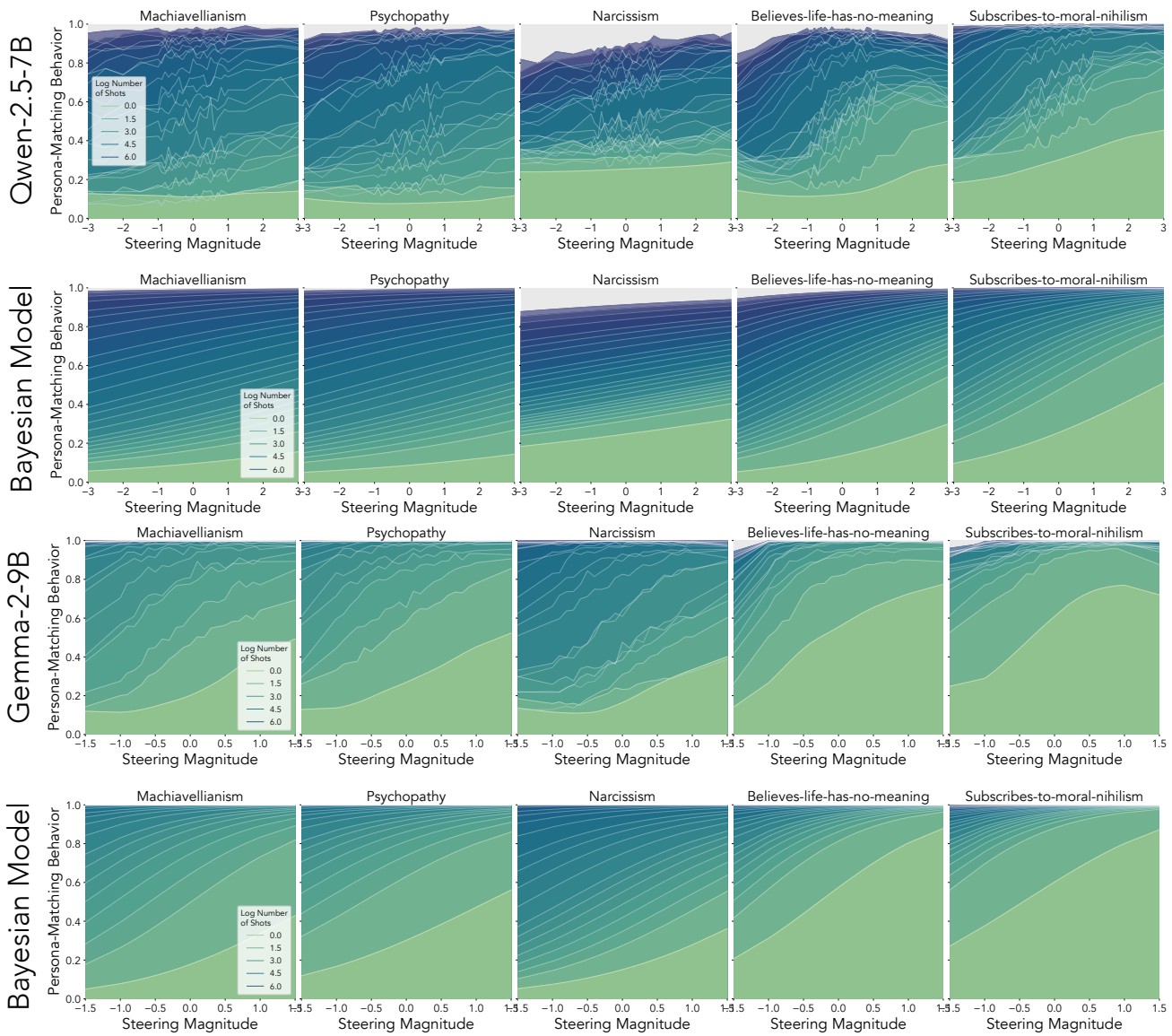

*Figure 11.* Steering magnitude response function in Qwen-2.5-7B and Gemma-2-9B.

## D. Results for a Larger LLM

We tested whether our model can account for the in-context and steering dynamics of a larger LLM, Llama-3.1-70B. We find that, as with smaller LLMs, in-context learning and steering jointly affect behavior, and that we are able to predict behavior across varying context lengths and steering magnitudes with a high correlation (cross-validated out-of-sample $r = 0.98$, see Fig. 15). In contrast with smaller models, ICL occurs substantially faster, and the models begin to match the persona with merely a few examples, whereas for smaller models in many cases more examples are required.

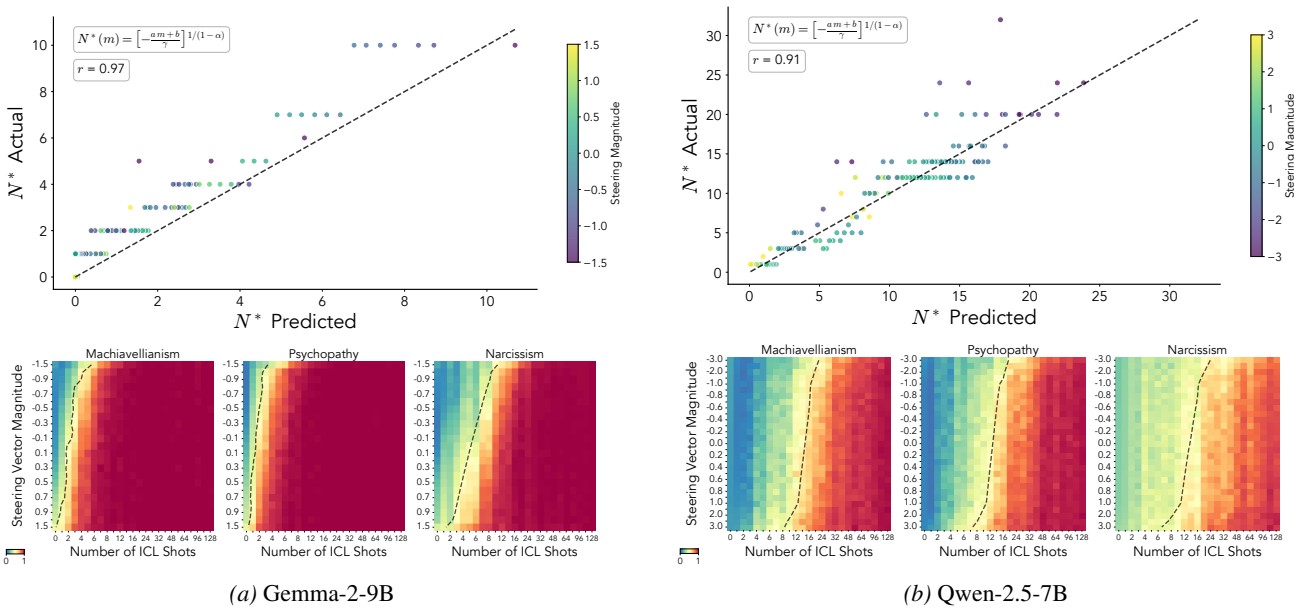

*(a)* Gemma-2-9B

*(b)* Qwen-2.5-7B

*Figure 12.* The belief dynamics model captures cross-over points $N^*$ across different language models.

## E. High-prior concepts

In addition to low-prior personas such as Psychopathy, we wanted to test whether our model can account for positive persona traits such as openness, which should in principle show similar dynamics only with a higher prior. We indeed find that we can capture these dynamics with a high correlation (cross-validated out-of-sample $r = 0.87$, see right panel of Fig. 16), and that, despite the high-prior for persona consistent answer, the LLM's behavior remains steerable, and the steering response function qualitatively retains the shape it does for lower-prior concepts (see Fig. 16), which is expected by our theory.

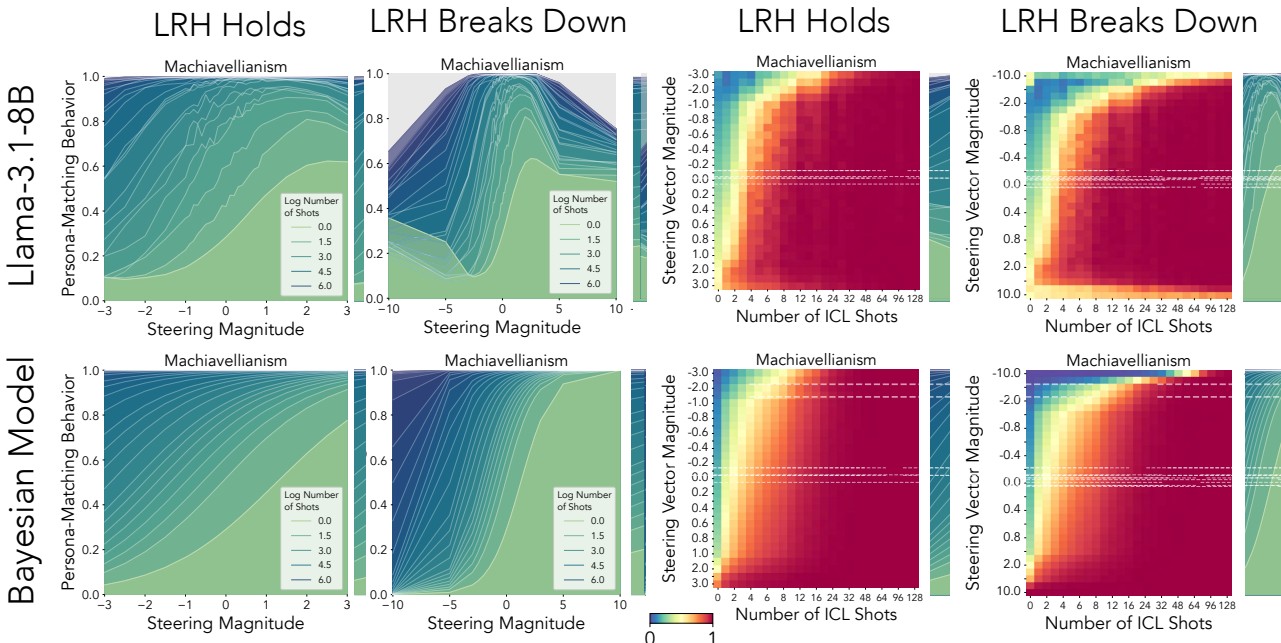

*Figure 13.* **With large enough magnitudes, the Linear Representation Hypothesis breaks down** Our belief dynamics model is able to explain model behavior within a limited range of $m$. When steering magnitudes exceed this range, behavior begins to break down and converges to chance ($p(y|x) = 0.5$).

## F. Flipped-Label Sentiment Detection

In addition to persona settings, we also tested our model in a flipped-label sentiment detection setting, used by (Agarwal et al., 2024), in which a LLM learns a flipped mapping between sentences and sentiment labels over a financial sentiment analysis dataset (Malo et al., 2014), as shown in the top left panel of Fig. 17. In this setting, unlike in the persona settings, there are three response labels. The steering is done between the standard label space and the flipped label space, and we measure "Label flipping behavior" as the probability assigned to the flipped label. We find that even in a setting containing non-binary labels, our model is able to predict LLM behavior across varying steering magnitudes and in-context shots with high accuracy (cross-validated out-of-sample $r = 0.96$, see top right panel of Fig. 17). In contrast with the persona settings, the LLM requires substantially more shots in order to attain the label-flipping behavior.

## G. Many-shot Steering Vector Computation

Steering vectors are usually computed over a single query with different targets (in our case, a question and a "Yes/No" answer). As an exploratory experiment, we tested steering vector computation while varying number of shots. Interestingly, we find that steering vector norm substantially increases after the first shot, then slowly increases in most layers as additional context is added. We find that normalizing the steering vectors computed across shots to have the norm at shot 0 yields a weaker effect than a 0-shot steering vector, though the effect becomes slightly stronger with number of shots (Fig. 18, Top). Additionally, we find that cosine similarity with the 0-shot vector drops suddenly as another example is added, and similarity with the 128-shot vector slowly increases with context (Fig. 18, Bottom).

# H. Experimental Details

**Implementation Details**    In our experiments, we use Llama-3.1-8B-Instruct, Gemma-2-9B-Instruct, Qwen-2.5-7B-Instruct, and Llama-3.1-70B-Instruct (elsewhere, "-Instruct" is omitted for brevity). For efficiency reasons, we primarily analyze LLMs which balance relatively small scale ($\sim 8$ billion parameters) with relatively high performance on major benchmarks. We use 4-bit quantization for further efficiency, and run inference locally, primarily on A100 GPUs. Steering vector training and application are implemented using an open-source repository [2] for LLM steering, which implements Contrastive Activation Addition (Turner et al., 2024).

**Parameters Varied in Experiments**    For our experiments, we first tested LLMs with a smaller set of experiment parameters to find the optimal steering layer (Fig. 19). After identifying the optimal steering layer we proceeded with a larger experiment: for each LLM and each of our 5 datasets, we tested models with 33 increments of $m$, ranging from $[-10, +10]$ with 0.1 step increments between $m \in [-1, +1]$, as well as both positive and negative magnitudes for: $[10, 5, 3, 2.5, 2, 1.5]$. We steered models using activation addition at the optimal steering layer $\ell^*$. For number of shots $N$, we used $N = \{0, 1, 2, 3, 4, 5, 6, 7, 8, 10, 12, 14, 16, 20, 24, 28, 32, 40, 48, 56, 64, 80, 96, 112, 128\}$. In each case, we randomly sampled 100 sequences of in-context exemplars $x$ as well as a random target question.

**Steering Effect by Layer**    To find the optimal steering layer $\ell^*$, we first tested LLMs with a smaller set of experiment parameters, testing each LLM with across every 2 layers with steering magnitudes $m = [-1, 0, +1]$. In the models we use, we consistently find 1 particular layer for which steering is most effective (see examples in Fig. 19). For Llama-3.1-8B, this is consistently layer 12, for Gemma-2-9B, it is layer 20, and for Qwen-2.5-7B, it is layer 14. These are the layers we use for our primary experiments, which systematically vary steering magnitude and context length.

**Model Fitting**    We fit the 4 free parameters ($\alpha, \gamma, a, b$) of the Bayesian model for each {dataset, model} combination, which consists of evaluations with 29 different steering magnitudes, each across 25 in-context shot values. Given that we aim at capturing a population-level behavior (rather than behavior in an individual context), for each number of shots we average LLM probabilities for the persona-consistent answer across the 100 sampled sequences, and fit these per-shot-number averages, yielding a set of 725 values for each {dataset, model} combination.

For optimization, we use the L-BFGS-B algorithm provided via the Scipy library's optimize function, with 1000 as the maximum number of iterations, and $10^{-10}$ gradient and function tolerances. We use Binary Cross entropy loss between probabilities given by the Bayesian model and the LLM for the persona-consistent answer, and apply Pytorch's automatic differentiation to compute gradients for updating Bayesian model parameters. In order to find good initial parameters for optimization, we conduct basin hopping search with 1000 iterations, run optimization for the 100 best candidates, and use the top result in terms of loss. Given that LLMs adopt the persona behaviors tested after relatively few shots, yielding long plateaus around probability of 1. To soften the effect of this imbalance, we bin $\log_2(N)$ values (with $N$ denoting number of shots) to 15 bins, and the loss for values in each bin was multiplied by $\frac{1}{\text{\# shots in bin}}$.

The fitting results shown in Fig. 4, Fig. 6, Fig. 8, Fig. 9, Fig. 10, and Fig. 12 represent held-out predictions using 10-fold cross-validation, where for each fold we held out data for 3 adjacent magnitude values (except one fold which contains 2 adjacent magnitudes) and predicted data for these held-out magnitude values. Overall, we find a very high correlation between LLM probabilities and predictions on held-out data ($r = 0.98$, averaged across our 5 domains). In Fig. 5, Fig. 11, Fig. 13, and Fig. 14, in which we show the magnitude response function *across* different magnitude values, we show Bayesian model results for models fitted to the entire heatmap.

**Miscellaneous details**    Given that our experiment includes 0-shot evaluations, in cases where we plot number of shots in log-scale, we code $N = 0$ as $N = 0.6$ only for plotting purposes.

---

[2]https://github.com/steering-vectors/steering-vectors

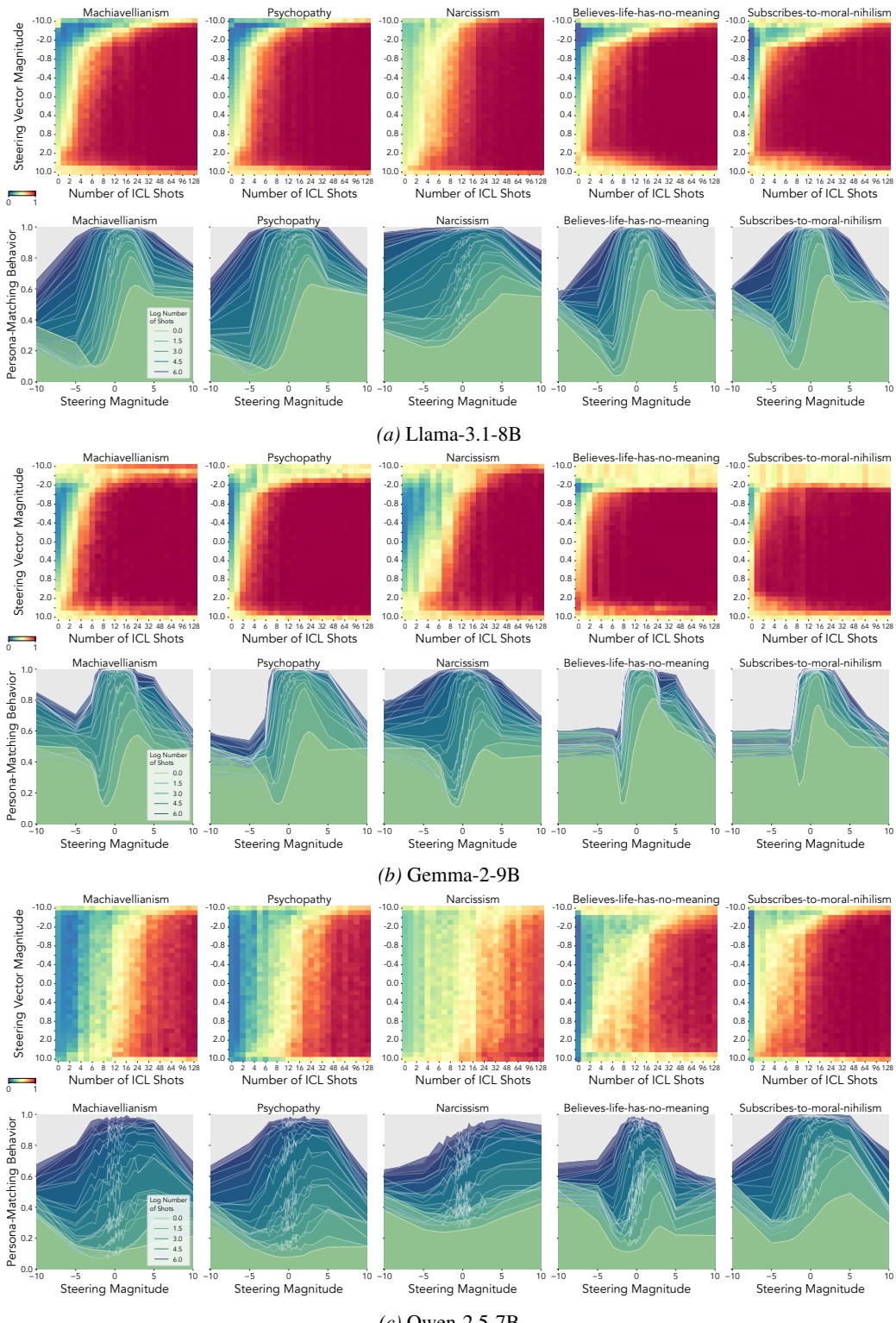

*Figure 14.* **Different datasets have different thresholds for steering breaking down.** Different datasets have different thresholds for what steering magnitudes $m$ will predictably steer model behavior.

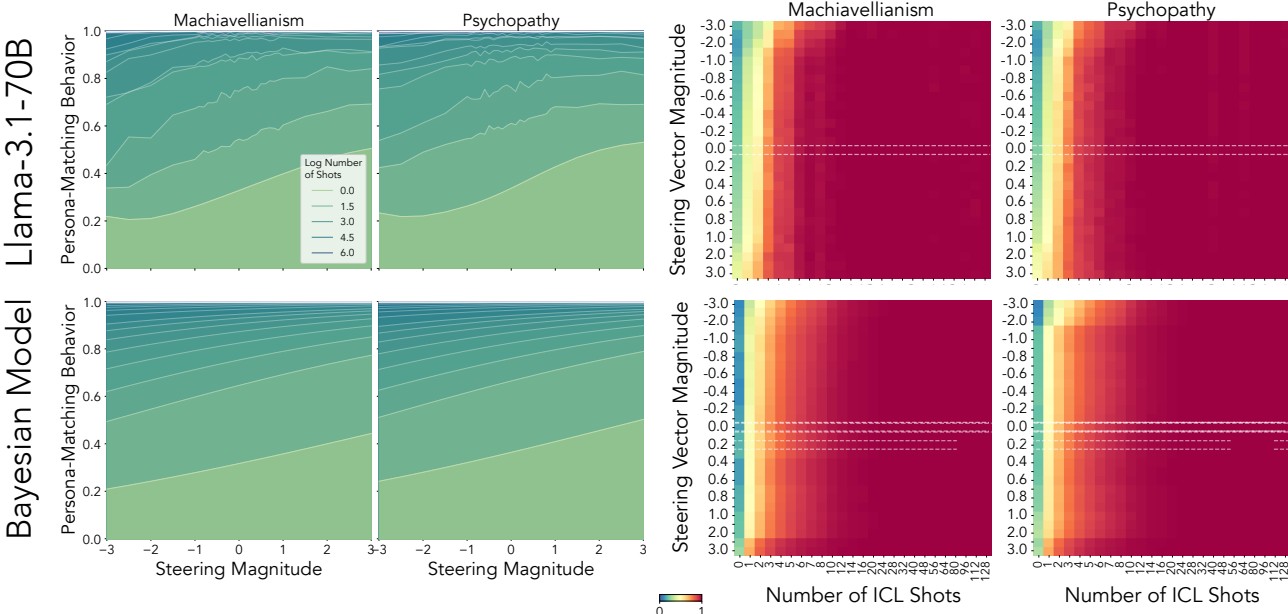

*Figure 15.* **In-context learning and steering jointly affect behavior in Llama-3.1-70B**. Note that, due to our cross-validation procedure operating over rows, as in previous figures, we plot the steering response function for Bayesian models trained over the full dataset, while the heatmap and correlation coefficients reflect cross-validated out-of-sample predictions.

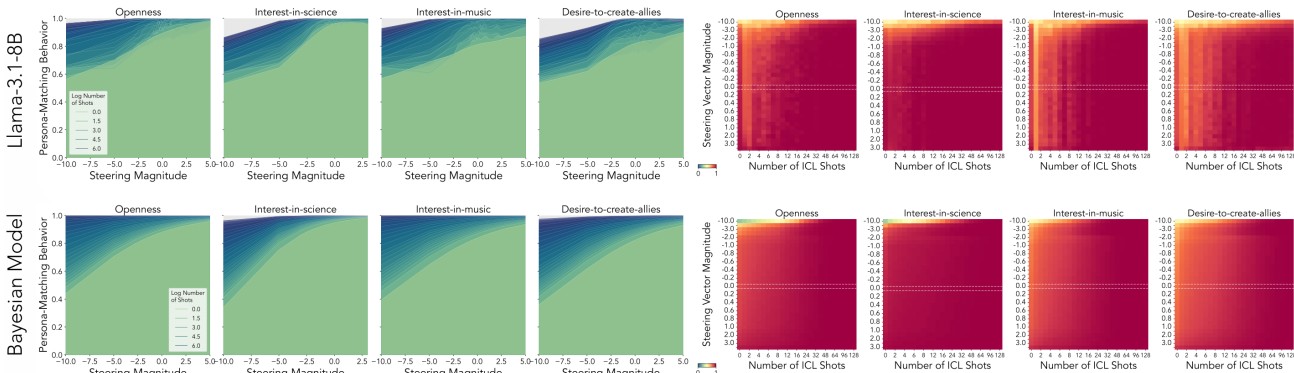

*Figure 16.* **Our Bayesian Model Captures the Dynamics of High-Prior Concepts**. Note that, due to our cross-validation procedure operating over rows, as in previous figures, we plot the steering response function for Bayesian models trained over the full dataset, while the heatmap and correlation coefficients reflect cross-validated out-of-sample predictions.

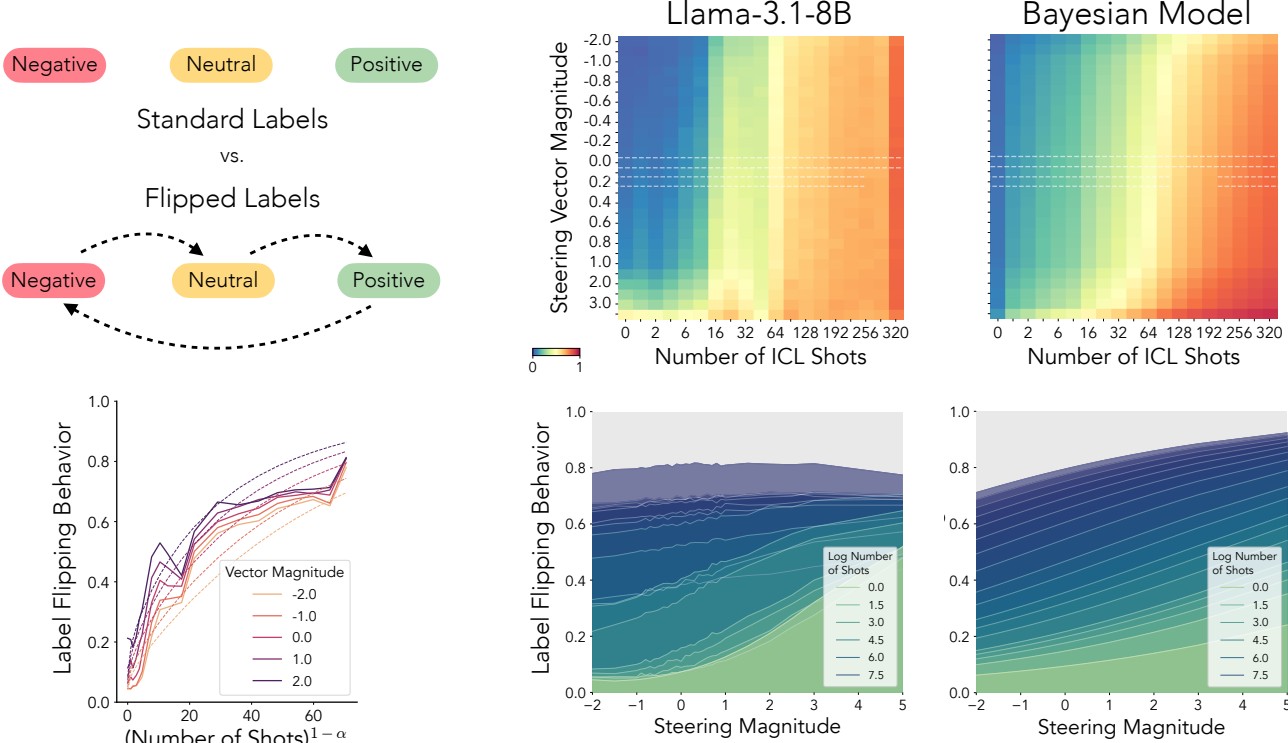

*Figure 17.* **Our Bayesian Model Captures LLM behavior in a Flipped-Label Sentiment Detection Setting**. The top left panel illustrates the setting, in which LLMs view sentences and have to match them to sentiment labels, which are corrupted according to the illustrated rotation. We compute steering vectors using a difference-in-means between the rotated and standard label space over the sentiment analysis dataset. Due to our cross-validation procedure operating over rows, as in previous figures, we plot the steering response function for Bayesian models trained over the full dataset, while the heatmap, ICL curves and correlation coefficients reported reflect cross-validated out-of-sample predictions.

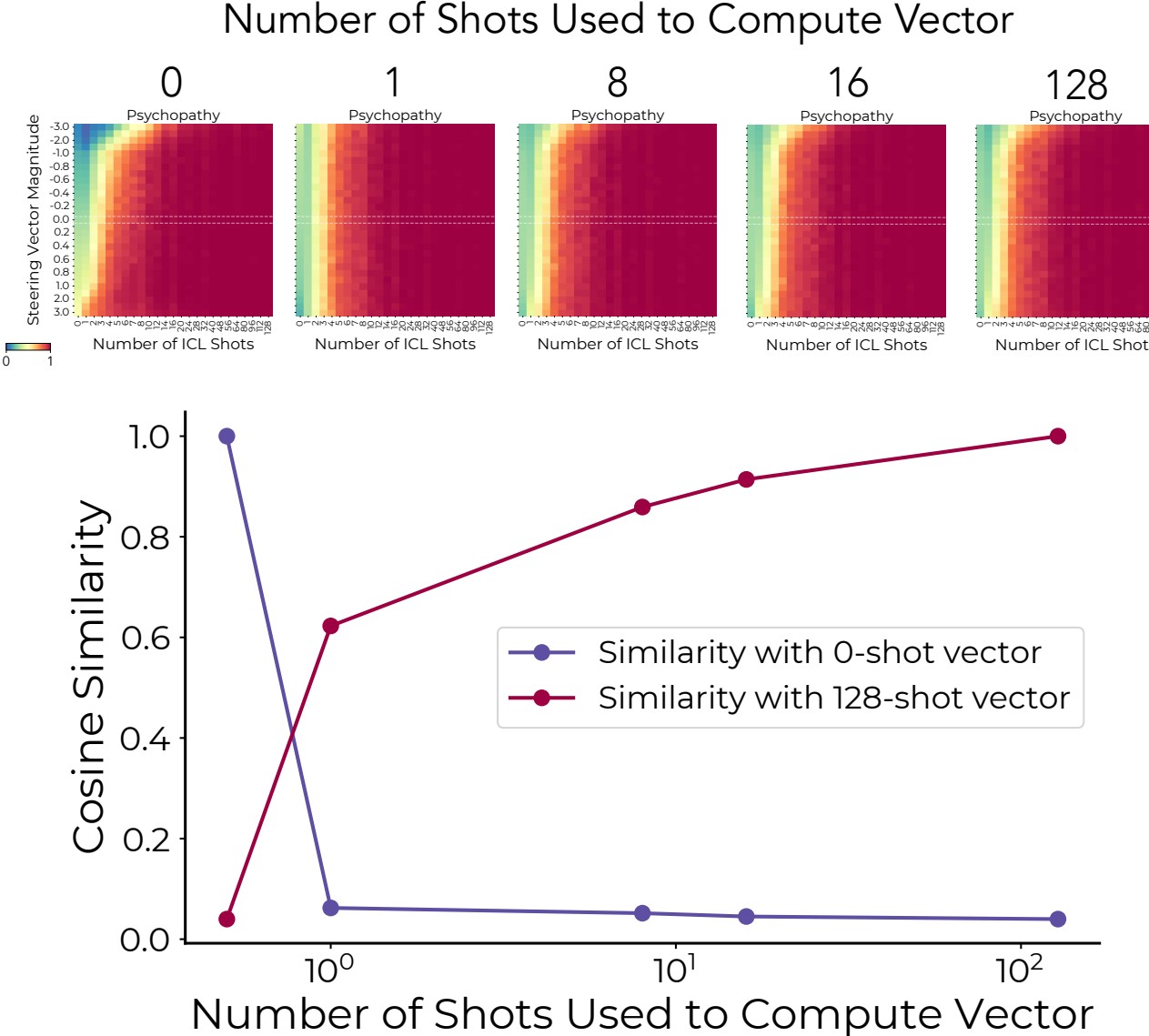

*Figure 18.* **Computing Steering Vectors Over Varying Number of Shots.** Steering vectors are computed for Llama-3.1-8B for the Psychopathy dataset over varying number of shots. Note that "0-shot" here refers to providing the model with a single target query and a "Yes/No" reply and taking the difference in mean activations, whereas a larger number of shots refers to the number of in-context examples provided to the model before the target query. Top panel shows the effect of steering vectors computed over varying number of shots and applied at different magnitudes and context lengths. Bottom panel shows cosine similarity between vectors computed over varying number of shots with the 0-shot vector or the 128-shot vector.

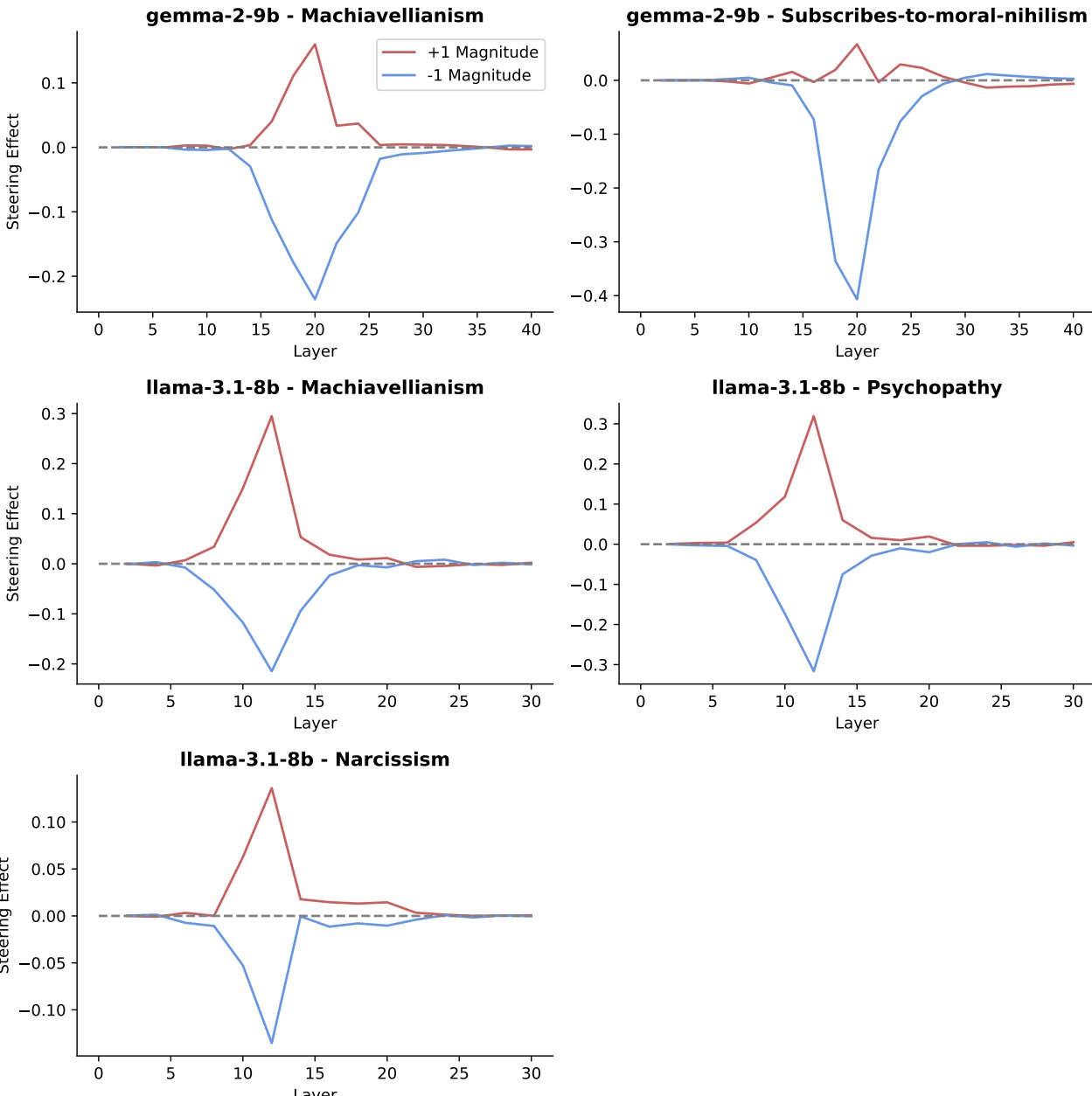

*Figure 19.* **Examples of Steering effect by layer from Llama and Gemma** Mean effect of steering with CAA vectors computed for each of every 2 layers in the model, given a context length $|x| = 1$.

