# OpenReview forum: "Belief Dynamics Reveal the Dual Nature of In-Context Learning and Activation Steering"
_ICML.cc/2026/Conference — ICML 2026 regular_

### Official Review · Reviewer_pvnb · 2026-03-08

**Soundness:** 2
**Presentation:** 2
**Significance:** 3
**Originality:** 3
**Overall Recommendation:** 4
**Confidence:** 3

**Summary:**

This paper tries to model the effect of few-shot demonstrations (i.e., ICL) and activation steering (i.e., task vectors) towards output into one mathematical model. The model describe the dynamics of ICL or activation steering as a function of output tendency against the ICL demonstration numbers and the activation steering magnitude.

**Compliance With Llm Reviewing Policy:**

Affirmed.

**Final Justification:**

Thanks for the author's reply. My concerns, except W2 has been addressed.

**Key Questions For Authors:**

1. Finding datasets whose inference accuracy can be significantly improved as datasets "correspond to concepts that LLMs assign relatively low probability to":

    - Using accuracy-like metric (the rate of yes/no matching) to evaluate such “improvement“ is also questionable. As we know, accuracy is not a linear metric as mentioned in previous works, and it may cause the S-shaped curve shown in Fig. 4 fake. How about utilizing the logits or cross-entropy loss of the expected answer?

    - Also, one more doubt is that: few-shot demonstrations may only change the expression of the answer, e.g., the LMs originally output “OK/sure“ but not “yes“, and when demonstrations are given, the LMs output “yes“, which can be observed as an accuracy improvement, and I think it is not related to some modification on “concept“. Can you exclude such a situation?

2. The $\alpha$ in Sec. 4.1 seems to be undefined in the main text. I apologize if I missed something.

**Limitations:**

Yes

**Strengths And Weaknesses:**

## Strength

- Jointly investigating ICL demonstrations and activation steering is highly interesting, since we already know that their effect is somehow similar.

- Incorporating the demonstration number and steering magnitude into the model is a good design. Many previous works missed such quantization, and therefore could not explain the well-known submodality of accuracy in ICL, but the model in this paper (as shown in Eq. 9) takes into account this problem.

- The experimental design is reasonable: Finding datasets whose inference accuracy can be significantly improved as datasets “correspond to concepts that LLMs assign relatively low probability to“ is appropriate, also, the selected task, i.e., transferring the LMs from positive personality to negative personality, matches intuition. Also, the flipped-label sentiment analysis task is also reasonable, however, I have some questions on this point. See question.

## Weakness

- I expected a clearer comparison of ICL and activation steering. As we know, the effects of activation steering and ICL demonstrations differ somewhat, e.g., in the accuracy curves as a function of ICL sample numbers or steering magnitudes. Currently this paper only claims that "steering operates by changing concept priors, while in-context learning leads to an accumulation of evidence", but no more explanation or evidence is provided to distinguish the difference between these two terms in detail.

- Based on the final model in Eq. 9, more interesting analysis can be made, e.g., analyzing the causality of parameters $a$, $b$, …, as some essential property against the model parameters or pre-training data, which is truly interesting in my opinion. The current paper seems to have ended abruptly at the most interesting point.

- Some results, e.g., Fig. 4 and 5, should be equipped with some significance analysis, e.g., the fitting goodness of your model against the experimental results.

---

> ### Author Rebuttal · Authors · 2026-03-31
>
> We are glad that the reviewer found our subject matter “highly interesting”, and that our model has “good design” and accounts for a factor that “many previous works missed”. We are also glad they find our experiment design to be “reasonable” and “appropriate” for our intended goals.
>
> Below, we respond to the Limitations and Questions specified by the reviewer:
>
> > I expected a clearer comparison of ICL and activation steering …
>
> Thank you for raising an important issue. Our initial motivation for this framing is that ICL operates as a function of input data $x$ and thus naturally maps onto a likelihood function $p(x | c)$ which depends on input data, whereas activation interventions directly alter model beliefs, and therefore intuitively maps onto a change in input-independent beliefs, i.e. priors $p(c)$. However, we also wanted to empirically test whether ICL and activation vectors operate by similar or different means, in terms of the representation changes they induce in LLMs.
>
> We investigated this question in Appendix G by computing steering vectors using varying numbers of ICL shots, and comparing these to (a) the single-shot (or “0-shot”) steering vectors which we used in the main text, and (b) vectors computed with 128 ICL shots. We find that even after including just one additional shot into the text, the vector is very different from the original steering vector, and conversely, with more ICL shots the vector becomes progressively more similar to the 128-shot ICL vector. This experiment provides initial evidence that ICL and CAA steering vectors operate by different means, since having more shots in-context makes the steering vector more distant and less similar.
>
>
>
> > Based on the final model in Eq. 9, more interesting analysis can be made, e.g., analyzing the causality of parameters $a$, $b$, …, as some essential property against the model parameters or pre-training data, which is truly interesting in my opinion  ...
>
> We agree that it is an interesting question of the parameters we infer with our model: what causes these particular values? How are these factors shaped by pre-training, fine-tuning, and details of model architecture? However, we believe that the scope of such an investigation would extend well beyond our core argument in this paper, and would be much more deserving of its own paper which explicitly focuses on this question. While this would provide important information about precisely why these components emerge, we do not think the absence of such an analysis detracts from our argument, since our claim and results will translate to other LLMs regardless of whether they have different concept priors $b$, steering sensitivity $a$, or likelihood functions ($\gamma$ and $\alpha$).
>
>
> > Some results, e.g., Fig. 4 and 5, should be equipped with some significance analysis, e.g., the fitting goodness of your model against the experimental results.
>
> Thank you for the suggestion, we agree that significance analysis is important to quantitatively demonstrate the validity of our model. We should clarify that the same data for LLM behaviors and model predictions is shown in Figures 4, 5, and 6, except that each of these plots projects different information onto the x- and y-axes. Therefore, showing significance in correlation between observed behaviors and model predictions once is sufficient to cover all three of these plots. On lines 204-207, we provide exactly this: “Overall, we find a very high correlation between LLM probabilities and predictions on held-out data (r= 0.98, averaged across our 5 domains; p<.001 for all correlations)”.
>
>
> > Finding datasets whose inference accuracy can be significantly improved as datasets "correspond to concepts that LLMs assign relatively low probability to" …
>
> We agree, matching of a single token yes/no would be misleading – this is precisely why we did exactly as you suggest! We measured the relative probabilities of tokens corresponding to “Yes” and “No”, including alternate capitalization such as “yes” or “YES” and single-letter responses “y” or “n”. We take the ratio of the summed probabilities for each “Yes” -matching token, and divide this by the sum of the probabilities for all “Yes” and “No” matching tokens.
> We agree that with the first shots of in-context learning the LLM may initially be learning that possible valid responses are “Yes” or “No”. However, since our method is comparing the relative ratios of these probabilities, we expect that initial uncertainty over answer format will not affect our main results.
>
> > The $\alpha$ in Sec. 4.1 seems to be undefined in the main text. I apologize if I missed something.
>
> Thank you for pointing this out. $\alpha \in [0, 1)$ is a scaling parameter which governs the particular rate at which beliefs are sub-linearly updated as a function of additional ICL examples. We will edit the text in Section 4.1 to explicitly define $\alpha$.

---

> > ### Author Rebuttal · Reviewer_pvnb · 2026-04-03
> >
> > Thanks for the author's reply. My concerns, except W2 has been addressed.
> >
> > I prefer to keep my current overall score to support your work.

---

### Official Review · Reviewer_rN85 · 2026-03-11

**Soundness:** 2
**Presentation:** 2
**Significance:** 2
**Originality:** 3
**Overall Recommendation:** 4
**Confidence:** 3

**Summary:**

This paper proposes a unified Bayesian account of two inference-time control mechanisms in LLMs: in-context learning and activation steering, and analyzes the concept of model control as belief updating over latent concepts. Based on this view, the paper derives a closed-form belief dynamics model that predicts three main phenomena. The paper validates these predictions primarily on persona-control tasks and a flipped-label sentiment task, and reports high held-out correlations between model predictions and observed LLM behavior across several models.

**Compliance With Llm Reviewing Policy:**

Affirmed.

**Final Justification:**

thanks for the responses and they have solved most of my comments except Q4. I would keep my current score.

**Key Questions For Authors:**

1. The paper argues that activation steering can be interpreted as changing prior odds, while ICL changes evidence through the likelihood term. What empirical evidence most directly distinguishes this decomposition from other possible parameterizations that could fit the same behavioral curves equally well?

2. Much of the main evaluation uses persona datasets where concept-consistent behavior is measured with relatively clean Yes/No outputs. How robust is the proposed account when the target behavior is less binary or less directly aligned with a single latent concept?

3. The derivation of the steering effect relies on the Linear Representation Hypothesis and approximate orthogonality of concept directions. Can the authors clarify which empirical findings in the paper most strongly support this assumption, as opposed to merely being consistent with it?

4. The model is fit to averages of behavior across sampled contexts rather than to raw per-context data. Do the authors have a sense of how much variability exists at the individual-context level, and whether the same theory can explain that variability rather than only the population average?

5. The paper reports that steering effects break down at sufficiently large magnitudes and that some models show weak or absent steering effects. How should these failures be interpreted relative to the proposed Bayesian account?

6. N* is one of the most interesting outputs of the model. How stable are these estimates across different random selections of in-context examples, prompt formatting choices, or answer-generation templates?

**Limitations:**

yes

**Strengths And Weaknesses:**

Strengths: The paper tackles an interesting and timely question at the intersection of ICL, activation steering, and mechanistic interpretability. Its main strength is the unifying perspective: instead of studying prompting and steering separately, it offers a compact Bayesian model that connects both through log-posterior belief updates. Besides, the paper does not just claim qualitative similarity, but derives explicit functional forms and cross-over points, then compares them against held-out behavioral data.

Weaknesses: The paper’s empirical fit is stronger than its causal validation. The central interpretation depends on several strong assumptions: that many-shot ICL can be modeled through a particular likelihood form with sublinear accumulation, and that activation steering operates by shifting prior odds under the Linear Representation Hypothesis. These assumptions are plausible and well motivated, but the paper mostly shows that the resulting model fits aggregate behavior, not that the underlying Bayesian decomposition is uniquely supported. Relatedly, the evaluation is concentrated on highly structured persona-control settings where the target concept and output behavior are especially cleanly aligned. This makes the analysis elegant, but it leaves open how well the same account extends to richer, less binary behaviors or more realistic prompting settings. Finally, the paper itself notes important caveats: steering breaks down at large magnitudes, some models show weak or unclear steering effects, and the model is fit to averages rather than individual contexts.

---

> ### Author Rebuttal · Authors · 2026-03-31
>
> We thank the reviewer for their feedback and are glad they appreciated the paper’s “interesting and timely” question, its “unifying perspective”, and that the paper does not merely claim qualitative similarity. We respond to specific comments below.
>
> > The paper’s empirical fit is stronger than its causal validation... What empirical evidence most directly distinguishes the decomposition...?
>
> We agree that our paper is stronger as a predictive account than as a proof of unique causal identifiability, and we will revise the paper to make this scope clearer. Our goal is not to claim that no other parameterization could fit some subset of the curves, but rather that this decomposition gives a simple and unified account that explains three distinct findings with a single additive log-odds model: (i) sigmoidal ICL dynamics as a function of $N^{1-\alpha}$ (Fig. 4), (ii) a sigmoidal steering response function with an approximately linear effect in log-odds space over moderate steering magnitudes (Fig. 5), and (iii) the joint phase boundary and crossover point $N^*$ induced by combining the two interventions (Figs. 6 and 8).
>
> > The evaluation is concentrated on highly structured persona-control settings... when the target behavior is less binary or less directly aligned with a single latent concept?
>
> We agree that the persona settings are especially clean, but note that we chose them for the clear practical use-case headlined by the Many-Shot Jailbreaking paper by Anil et al. More critically, we highlight we did include an additional task beyond binary persona control: flipped-label sentiment analysis (Fig. 7, App. F). In this setting, the model must learn a three-way permuted label mapping [Negative, Neutral, Positive] $\to$ [Positive, Negative, Neutral]. We still find that our model predicts behavior well across steering magnitudes and in-context shots (cross-validated out-of-sample $r^2$=0.96)! While richer behaviors may not align neatly with a single binary latent concept, we view this experiment as an important first step beyond the cleanest binary persona setting.
>
> > The derivation of the steering effect relies on LRH... rather than merely being consistent with it?
>
> The strongest evidence for this in our paper is that steering magnitude produces the specific response function predicted by Eq. 8: behavior changes sigmoidally in probability space and approximately linearly in log-odds space over a moderate range of magnitudes, and this pattern holds across multiple context lengths (Fig. 5). In addition, in App. H / Fig. 19, we find that steering is typically concentrated in one particular layer (or a small number of nearby layers), which is consistent with the idea that belief in these concepts is both linearly accessible and localized.
>
> > The model is fit to averages across sampled contexts... variability at the individual-context level?
>
> Our current model is indeed intended as a population-level account: as described in the App. H, for each condition we sample 100 random in-context sequences (and a random target question), average the persona-consistent probabilities across those sequences, and then fit the model to those averages. That said, we qualitatively saw the individual samples show a transition in behavior generally near the population phase boundary. To better quantify and demonstrate this, we promise to add a plot of variance w.r.t. the aggregate dynamics in the final paper. Thank you for this suggestion!
>
> > The paper reports that steering effects break down at sufficiently large magnitudes... interpreted relative to the proposed Bayesian account?
>
> We view these cases as important boundary conditions on the account, rather than contradictions to it. Our main theory is explicitly about the regime where steering changes posterior belief monotonically. In App. C / Fig. 13, when steering magnitudes become too large, representations are pushed outside the regime where the linear approximation is valid, and behavior breaks down toward chance. Similarly, as we note in the Discussion, weak or absent steering effects in some models may indicate that the relevant concept is not linearly represented in a usable way or that there is insufficient signal for that concept in that model. We will revise the paper to emphasize this point further!
>
> > $N^*$  is one of the most interesting outputs of the model. How stable are these estimates across different random...?
>
> In the current paper, stability across random selections of in-context examples is addressed indirectly by the experimental design: each condition averages over 100 randomly sampled in-context sequences, and despite that variability we find that predicted and empirical crossover points are highly aligned (Fig. 8, r=0.97). However, we did not explicitly perform a dedicated ablation over prompt formatting or answer-generation templates; as noted above, we will add a plot of deviation from predicted $N^*$ to quantitatively address this in the final paper!

---

> > ### Author Rebuttal · Reviewer_rN85 · 2026-04-03
> >
> > thanks for the responses and they have solved most of my comments except Q4. I would keep my current score.

---

### Official Review · Reviewer_896k · 2026-03-12

**Soundness:** 3
**Presentation:** 2
**Significance:** 3
**Originality:** 3
**Overall Recommendation:** 4
**Confidence:** 1

**Summary:**

This paper introduces a unifying Bayesian framework to characterize two distinct control mechanisms—In-Context Learning (ICL) and activation steering—both as manifestations of a single underlying process. The authors propose that both interventions update model beliefs about latent concepts: activation steering shifts concept priors, while ICL provides evidence through input examples to update the posterior according to likelihood functions.

The framework successfully explains key behaviors of large language models (LLMs), such as the sigmoidal learning curves observed in many-shot ICL, and predicts that both interventions are additive within the log-belief space. This additivity explains how small changes in intervention conditions can lead to sudden, dramatic shifts in model behavior, such as those seen in many-shot jailbreaking. The authors empirically validate the theory using Contrastive Activation Addition (CAA) on models like Llama-3.1-8B, Qwen-2.5-7B, and Gemma-2-9B across domains like personality traits and sentiment analysis, achieving strong alignment between empirical results and Bayesian predictions (r=0.98). Overall, this work provides a unified and quantifiable framework for understanding and predicting LLM behavior, supported by both theoretical and experimental evidence.

**Compliance With Llm Reviewing Policy:**

Affirmed.

**Final Justification:**

I would like to maintain the current score to support this solid work.

**Key Questions For Authors:**

I have no questions for the authors. Please refer to other reviewers' questions.

**Limitations:**

1. The paper misses the Impact Statement section, which is required.
2. The assumption that concepts in neural network representations follow the Linear Representation Hypothesis works well in certain tasks. However, when handling more complex or nonlinear concepts, the model may not fully capture these relationships, which could limit its broader applicability.

**Strengths And Weaknesses:**

Strengths:
1. The paper unifies ICL and activation steering within a Bayesian framework, modeling ICL as evidence accumulation and activation steering as prior intervention. The experimental design is thorough, testing multiple tasks and variables, with results closely matching predictions.
2. The paper is well-structured and clearly written, effectively positioning its contributions within the literature and explaining its methodology.
3. The paper addresses a key issue in large model security, offering a novel framework that predicts behavior changes and opens new directions for more secure, controllable models.

Weaknesses:
1. No Impact Statement provided.
2. The discussion section is quite long, and while thorough, it could benefit from being more concise. Breaking it into smaller, more digestible sections would help readers focus on key insights and make the paper easier to follow.

---

> ### Author Rebuttal · Authors · 2026-03-31
>
> We are glad that the reviewer appreciated our theory which unifies ICL and activation steering, and that they found our experiments to be “thorough”. We are further grateful for their comments that the paper is “well-structured and clearly written”, and that it “addresses a key issue” with a “novel framework” that “opens new directions”.
>
> Below, we respond to the Weaknesses and Limitations raised by the reviewer:
>
> > No Impact Statement provided.
>
> Unfortunately we realized this only after the paper had been submitted. We will update our submission with the following impact statement:
>
> *This work studies how large language models (LLMs) can be controlled by in-context learning and activation steering. By modeling these interventions as Bayesian belief updates, our approach may enable more principled and reliable methods for steering LLMs through both behavior (prompting) and activation interventions. Improved model steering could have important implications for AI safety, for example mitigating risks of jailbreaks or unintended persona adoption. However, these insights could potentially be misused to more effectively manipulate LLMs or bypass safety mechanisms.*
>
>
> > The discussion section is quite long, and while thorough, it could benefit from being more concise. Breaking it into smaller, more digestible sections would help readers focus on key insights and make the paper easier to follow.
>
> Thank you for the suggestion. Accordingly, we will edit our Discussion section to be more concise and readable, and we will add paragraph headers such as “Limitations” and “Future Work”.
>
>
>
> > The assumption that concepts in neural network representations follow the Linear Representation Hypothesis works well in certain tasks. However, when handling more complex or nonlinear concepts, the model may not fully capture these relationships, which could limit its broader applicability.
>
> We agree, the assumption of linearity is a limitation of our work, and one that we hope to overcome in follow-up work. We hope to explore more complex and non-linear concepts in later work, which would require novel activation steering methodologies.
>
> In order to begin addressing this point, we included the additional domain of flipped-label sentiment analysis (Figure 7, App. F) which is more complex and arguably non-linear in nature. In this task, the original sentiment labels [Negative, Neutral, Positive] are permuted to the ordering [Positive, Negative, Neutral]. Permutation is a concept which is modular and cyclical, so in principle it could be considered non-linear, although, since we find success with activation steering, this suggests some degree of linearity in representation.

---

> > ### Author Rebuttal · Reviewer_896k · 2026-04-03
> >
> > Thanks for your feedback. I already gave a decent score to this paper, so I will maintain my score. Good luck!

---

### Official Review · Reviewer_9gES · 2026-03-16

**Soundness:** 3
**Presentation:** 3
**Significance:** 2
**Originality:** 2
**Overall Recommendation:** 4
**Confidence:** 4

**Summary:**

The paper tries to unify prior work on in-context learning and activation steering through what they call a belief dynamics model built on a Bayesian inference framework. The work builds on existing literature that models the effect of increasing in-context examples as updating concept posteriors (specifically, the likelihood) in Bayesian inference. This work presents a unifying framework under the linear representation hypothesis, additionally modeling the effect of activation steering as updating beliefs through concept priors. This is validated by experiments mainly using Llama, and also some with Qwen and Gemma models on persona-matching and flipped-label sentiment analysis datasets.

**Compliance With Llm Reviewing Policy:**

Affirmed.

**Final Justification:**

The authors agreed to fix discussion on a prior work, a sign error in their analysis, small presentation things I pointed out in the review.

I was initially a little skeptical about the significance and originality of the work, the authors provided a nice contextualization of their results wrt to prior work in their rebuttal.

While the work borrows and builds on many small ideas from prior work, providing a unified dual-mechanism picture of ICL and steering is a nice and clean result, a good formal account that may seem easy to follow in hindsight, but is nonetheless valuable to have out there. Hence, I recommend a weak accept.

**Key Questions For Authors:**

Please see the strengths and weaknesses section.

**Limitations:**

yes

**Strengths And Weaknesses:**

**Strengths**

The paper is well-written and easy to follow. The authors present and discuss prior literature and their results with good clarity, and the cartoon figures are neat and effectively illustrate their points. Second, the topic is highly relevant given the large literature on ICL and activation steering, so this paper would be of interest to a relatively large portion of the community. Further, the work presents a nice unifying account of ICL and activation steering by combining previous literature on ICL and formalizing steering under Linear Representation Hypothesis in a Bayesian inference model.

The paper also includes a good discussion section on broad challenges of activation steering, such as locality and potential non-linearity in the context of this work, and acknowledges some limitations of the work, such as only working with binary concepts.

**Weaknesses**

Given the large existing literature on these topics, the unification of ICL and activation steering, while seemingly sound, I am not so sure about its significance and originality. First, as the paper acknowledges, ICL as Bayesian inference has been significantly explored, and prior literature already establishes that increasing in-context examples pushes the model posterior. This part of the work non-trivially builds on prior work: Anil et al. (2024) already captures a sharp learning curve as a function of $N$ in the context of jailbreaking, and Wurgaft et al. (2025) captures the sub-linear effect of $N$ on the log-likelihood via the $N^{-\alpha}$ scaling factor. The ICL section therefore seems fairly incremental.

Turning to activation steering, which feels like the bigger potential contribution of this work, it reads more as a good formalization of what already follows from prior work than a significant new result. Intuitively, if the Linear Representation Hypothesis holds and one assumes a Bayesian inference model, it follows naturally that steering would update beliefs without affecting the log-likelihood term. While it is nice to see this formalized, and that one can predict the threshold $N_∗$ that enables a persona, these contributions don't seem significant.

Section 2.1. The paper cites Raventós et al. (2024) and Park et al. (2024a) as "verifying that different perspectives with ICL can be captured by casting it as Bayesian inference". If I am not mistaken, Raventós et al. (2024) specifically show that the generalization regime is non-Bayesian (or equivalently, Bayesian inference over the full task distribution as opposed to just the pre-training task distribution). Park et al. (2024a) similarly identify qualitative phase transitions between different forms of retrieval and learning algorithms that resist a uniform Bayesian account. This is a subtle but important distinction, which I suggest should be addressed.

Section 4.2. There appears to be a sign error in the calculation of the posterior odds. Given Eq. (7), the log posterior odds should be  $-a\beta_i(v) - b$. I believe the authors intended to define Eq. (7) as $\sigma(w^\top v + b)$ instead of $\sigma(-w^\top v - b)$, the latter saturates $p(c|v)$ to 0 rather than 1 for large $\beta_i(v)$, which is the opposite of the desired behavior. Similarly, Eq. 3 derives posterior in terms of sigmoid of posterior-odds, Eq. 5 however, uses a minus sign in the expression for some reason, which gives the opposite effect of increasing $N$ and prior odds. Let me know if I am missing something.


Section 2.2, the CAA paragraph, defines concept vectors as directions between ordered pairs such as (male, female) or (king, queen). However, the rest of the paper, treats each concept as having its own independent vector. For instance, the formulation of LRH in Eq. 6. I believe one could assume that male and female are simply flipped versions of one another, but given the writing in Section 2.2, this should be clarified.

Small points: Section 2.2 please mention $m$ is steering strength. Please mention $d_{c, \ell}$ is the concept vector that is used to steer for/against concept $c$.

---

> ### Author Rebuttal · Authors · 2026-03-31
>
> We thank the reviewer for the detailed and thorough review! We are glad that the reviewer appreciated our work, noting that it presents a “nice unifying account of ICL and activation steering” and stating that the paper will be “of interest to a relatively large portion of the community.” Furthermore, we are grateful that the reviewer found our paper’s presentation quality compelling, calling it “well-written and easy to follow” and stating our demonstration figures were “neat” and “effectively illustrate” our points.
>
> Below, we respond to the Weaknesses and Limitations raised by the reviewer:
>
> > Given the large existing literature on these topics…
>
> We thank the reviewer for this comment. While Wurgaft et al. (2025) derive a sublinear scaling factor for the log-likelihood, their focus is on *training* dynamics, whereas ours is on *inference-time* belief dynamics. Anil et al. (2024) characterize sharp learning curves in many-shot jailbreaking but do not provide a Bayesian account. Our contribution is to bring these threads together into a single closed-form Bayesian model that is quantitatively predictive of LLM behavior across a range of settings. More broadly, we would push back on evaluating the ICL and steering components in isolation: the central contribution is precisely the *unification*, where both interventions operate by updating the same belief state additively in log-odds space, yielding concrete predictions about their joint effect --- including the phase boundary and $N^*$. To our knowledge, no prior work derives such a unified model and validates it with the quantitative fit we report across multiple models and domains.
>
> > Turning to activation steering, which feels like the bigger potential contribution of this work…
>
> We appreciate the reviewer's engagement with this point, though we would like to offer a slightly different perspective. While it may seem in hindsight that the effect of steering on prior odds follows naturally from LRH and a Bayesian model, assuming both frameworks hold *simultaneously* and interact in the specific additive way we describe is itself a non-trivial claim --- one could have imagined that steering interferes with the likelihood term, or that the two interventions interact non-linearly, neither of which is ruled out by prior work. The fact that our model achieves high quantitative fit across multiple models and domains (cross-validated $r = 0.98$) is therefore an empirical contribution that goes beyond formalization alone, and the ability to predict $N^*$ as a function of steering magnitude is a concrete novel upshot. Finally, we reiterate that the steering contribution is best evaluated in the context of the unified model --- it is the *joint* account of ICL and steering as dual mechanisms of belief updating, and its empirical validation, that constitutes the core contribution of this work.
>
> > Section 2.1. The paper cites Raventós et al. (2024) and Park et al. (2024a) as…
>
> We thank the reviewer for this nuanced point. We note that the reviewer themselves observe that Raventos et al.'s generalization regime can be interpreted as ``Bayesian inference over the full task distribution'' --- which is precisely the reconciliation we have in mind, and which Wurgaft et al. (2025) support empirically. Thus neither Raventos et al. nor Park et al. strictly contradict a Bayesian account of ICL, though we agree they contradict an account characterized by a single Bayesian predictor. We will correct the sentence in the revision to reflect this distinction.
>
> > Section 4.2. There appears to be a sign error in the calculation of the posterior odds…
>
> We are grateful to the reviewer for catching this. The issue is a notational inconsistency between the main text and the appendix: the appendix (A.2, A.3) implicitly uses a nonstandard decreasing sigmoid convention, within which its derivations are internally consistent, while Eq.3 uses the standard increasing sigmoid, causing Eq.5 and Eq.7 to inherit a spurious minus sign. Crucially, the final model (Eq.9) is a log-odds expression and is entirely unaffected, as are all empirical results. We will correct the sign error in the revision.
>
> > Section 2.2, the CAA paragraph…
>
> We thank the reviewer for this comment. Our treatment of concept vectors is consistent with Section 2.2: Figure 3 illustrates a concept vector as the direction between a target persona $c$ and a neutral persona $c'$, and Eq.6 posits a single direction capturing a binary concept rather than separate vectors for each pole, with $\beta_i(v)$ capturing the extent to which one pole is active. We will clarify this connection in the revision.
>
> > Small points: Section 2.2 please mention…
>
> We will address these in the revision, and apologize for the lack of clarity!

---

> > ### Author Rebuttal · Reviewer_9gES · 2026-04-04
> >
> > Thanks to the authors for the detailed response.
> >
> > Thanks for agreeing to address and fix the Raventos et al. citation point, sign error, etc.
> >
> > Other than that, thanks for offering a contextualization of the work with respect to Wurgaft et al. (2025) and Anil et al. (2024); on further reflection and reevaluation, I agree that while the work borrows and builds on many small ideas from prior work, providing a unified dual-mechanism picture with steering is a nice and clean result, a good formal account that may seem easy to follow in hindsight, but is nonetheless valuable to have out there. On that note, I will increase my score.

---

> > > ### Author Response · Authors · 2026-04-05
> > >
> > > Thank you very much for your reply, we greatly appreciate your willingness to engage with our response and reevaluate our work, and for your detailed and thorough review!

---

### Decision · Program_Chairs · 2026-04-30

**Decision:**

Accept (regular)

**Comment:**

This paper proposes a unified Bayesian framework to understand both in-context learning (ICL) and activation steering as forms of belief updating in language models. The authors explore the concept of model “belief” over latent concepts, arguing that ICL contributes evidence while activation steering shifts priors, and that both combine additively in log-odds space.

Reviewers agreed that the paper is clear, technically sound, and addresses an important and timely topic. A key strength is the unifying perspective: rather than treating ICL and steering as separate phenomena, the paper provides a simple model that explains known behaviors (such as sigmoidal ICL curves) and makes concrete predictions about how these interventions interact. The empirical results show strong agreement between the model and observed behavior across multiple models and tasks, which supports the usefulness of the framework.

The main concerns were about novelty and the strength of the claims. Some reviewers noted that parts of the framework build on existing ideas, and that the empirical validation is more predictive than causal, relying on assumptions such as linear representations and relatively clean, structured settings. After the rebuttal, these concerns were largely addressed. The authors clarified how their contribution differs from prior work, fixed technical issues, and better scoped their claims. Reviewers acknowledged these improvements, with one increasing their score and others marking concerns as resolved or mostly resolved.

From my own reading, I found the paper interesting and thought-provoking. The core idea is simple, but it offers a clean way to connect two widely used techniques and helps make sense of several empirical observations under a single lens. While the assumptions and evaluation scope limit how broadly the results can be interpreted, they do not undermine the main contribution. More importantly, the paper provides a useful conceptual framing that I expect will make people think and could inspire follow-up work.

I recommend acceptance. This is a solid, well-executed and well-written paper with a clear idea, strong empirical support, and a unifying perspective that is likely to be useful to the community working on interpretability and control of language models.